# Architecture of a mammalian glomerular domain revealed by novel volume electroporation using nanoengineered microelectrodes

D. Schwarz[1,2,3], M. Kollo[1,4,5], C. Bosch[4,5], C. Feinauer[1,3], I. Whiteley[4,5], T.W. Margrie[6], T. Cutforth[7] & A.T. Schaefer[1,3,4,5]

Dense microcircuit reconstruction techniques have begun to provide ultrafine insight into the architecture of small-scale networks. However, identifying the totality of cells belonging to such neuronal modules, the "inputs" and "outputs," remains a major challenge. Here, we present the development of nanoengineered electroporation microelectrodes (NEMs) for comprehensive manipulation of a substantial volume of neuronal tissue. Combining finite element modeling and focused ion beam milling, NEMs permit substantially higher stimulation intensities compared to conventional glass capillaries, allowing for larger volumes configurable to the geometry of the target circuit. We apply NEMs to achieve near-complete labeling of the neuronal network associated with a genetically identified olfactory glomerulus. This allows us to detect sparse higher-order features of the wiring architecture that are inaccessible to statistical labeling approaches. Thus, NEM labeling provides crucial complementary information to dense circuit reconstruction techniques. Relying solely on targeting an electrode to the region of interest and passive biophysical properties largely common across cell types, this can easily be employed anywhere in the CNS.

[1] Behavioural Neurophysiology, Max Planck Institute for Medical Research, Jahnstraße 29, Heidelberg, 69120, Germany. [2] Department of Neuroradiology, Heidelberg University Hospital, Im Neuenheimer Feld 400, Heidelberg, 69120, Germany. [3] Department of Anatomy and Cell Biology, Faculty of Medicine, University of Heidelberg, Im Neuenheimer Feld 307, Heidelberg, 69120, Germany. [4] Neurophysiology of Behaviour Laboratory, The Francis Crick Institute, 1 Midland Road, London, NW1 1AT, UK. [5] Department of Neuroscience, Physiology & Pharmacology, University College London, Gower Street, London, WC1E 6BT, UK. [6] The Sainsbury Wellcome Centre for Neural Circuits and Behaviour, University College London, 25 Howland Street, London, W1T 4JG, UK. [7] Department of Neurology, Columbia University Medical Center, 650 West 168th Street, New York, 10032 NY, USA. Correspondence and requests for materials should be addressed to D.S. (email: daniel.schwarz@med.uni-heidelberg.de) or to A.T.S. (email: andreas.schaefer@crick.ac.uk)

The interplay of convergent and divergent networks has emerged as one of the organizational principles of information processing in the brain[1]. Dense circuit reconstruction techniques have begun to provide an unprecedented amount of anatomical detail regarding local circuit architecture and synaptic anatomy for spatially limited neuronal modules[2–4]. These techniques, however, still rely predominantly on preselection of target structures, because the volumes that can be analyzed are generally small when compared to brain structures of interest (see, however, recent advances in whole-brain staining[5]), or remain confined to simpler model organisms[6,7]. Viral tracing approaches, on the other hand, depend on virus diffusion and tropism, thus infection probability is highly variable among different cell populations, preventing robust selection of a defined target volume[8,9]. Therefore, functionally dissecting a specific neural microcircuit, which typically extends >100 μm, and identifying its corresponding projections remains a challenge. The simultaneous requirement for completeness (i.e., every neuron in a target volume) and specificity (i.e., labeling restricted to neurons in a target volume), in particular, is challenging using current techniques.

Targeted electroporation as a versatile tool for the manipulation of cells was initially introduced as a single-cell approach[10], which was later proposed for delineating small neuronal ensembles using slightly increased stimulation currents[11]. It still remains the state-of-the-art technique for specific, spatially restricted circuit labeling and loading[12,13]. The exact spatial range and effectiveness of electroporation, however, remains poorly understood and is generally thought to be restricted to few micrometers[14]. In the brain, dedicated microcircuits are often engaged in specific computational tasks such as processing of sensory stimuli. These "modules" or "domains" are often arranged in stereotyped geometries, as is the case for columns in the barrel cortex[15] and spheroidal glomeruli in the olfactory bulb[16].

Here, we report the development of nanoengineered electroporation microelectrodes (NEMs), which grant a reliable and exhaustive volumetric manipulation of neuronal circuits to an extent >100 μm. We achieve such large volumes in a nondestructive manner by "gating" fractions of the total electroporation current through multiple openings around the tip end, identified by modeling based on the finite element method (FEM). Thus, a homogenous distribution of potential over the surface of the tip is created, ultimately leading to a larger effective electroporation volume with minimal damage. We apply this technique to a defined exemplary microcircuit, the olfactory bulb glomerulus, thereby allowing us to identify sparse, long-range and higher-order anatomical features that have heretofore been inaccessible to statistical labeling approaches.

## Results

### Evaluating efficacy of standard electroporation electrodes.
To provide a quantitative framework for neuronal network manipulation by electroporation, the volumetric range of effective electroporation was first calculated by FEM modeling; under standard conditions for a 1 μA electroporation current[10,14], the presumed electroporation threshold of 200 mV transmembrane potential[17] is already reached at approximately 0.3 μm distance from the tip, by far too low for an extended circuit (Fig. 1a, b). To achieve electroporation sufficient for such a volume, the stimulation current would have to be increased by a factor of 100, leading to an effective electroporation radius of more than 20 μm (Fig. 1c, d). At the same time, however, this would also substantially increase the volume experiencing >700 mV, which is thought to be the threshold for irreversible damage and lysis for many cellular structures[18]. Correspondingly, translating

these numbers to in vitro validation experiments shows the destructive nature of standard electroporation; increased stimulation intensity frequently results in "jet-like" convection movement and gas bubble formation. Both occur beyond a current threshold that scales with tip radius, and are notably within the range of currents needed to label even small neuronal circuits (Fig. 1e, f). Nevertheless, our modeling results were in excellent agreement with experimental measurements of the induced electric potential for a standard patch clamp setup (Supplementary Fig. 1).

### Design of nanoengineered electroporation microelectrodes.
To avoid these destructive forces during volume electroporation, the current density should be reduced while maintaining the overall electroporation volume. We again employed FEM modeling to determine a suitable distribution of current across the pipette surface. Using the axial resistance of the internal electrolyte as a current divider, this could in principle be achieved by inserting small, ~1–2 μm holes at varying distances from the tip and channeling fractions of the current through multiple release sites (Fig. 2a). Insertion of holes at only five distances results in a 50% decrease in maximum current density at a comparable overall electroporation volume to a standard pipette (Fig. 2b–d), whereas fewer and larger holes lead to substantial volume loss under the same conditions (Supplementary Fig. 2). Concomitantly, FEM shows that by using such an electrode design for electroporation, peak potentials are reduced by 80% and the volume at risk of irreversible damage (i.e., experiencing >700 mV transmembrane potential) is reduced tenfold for essentially the same overall electroporation volume (Fig. 2c, d). Indeed, for the same volume at risk of irreversible damage, volumes of more than 12-fold larger can be effectively labeled.

To construct such "nanoengineered electroporation microelectrodes" (NEM), we used focused ion beam (FIB)-assisted milling. Standard patch pipettes were pulled on a horizontal micropipette puller, sputter coated with gold and silver painted, in order to prevent charging in the scanning microscope. Following FIB, four rows of five holes were created, each set apart by 90° around the tip (Fig. 2e–g; for details, see "Methods" section).

### Effectiveness of NEM electroporation in vivo.
To assess the completeness and specificity of NEM electroporation, we labeled a typical medium-sized neuronal module in mice, the olfactory glomerulus (Fig. 3a–e). Two dyes were electroporated sequentially. First, a pipette loaded with tetramethylrhodamine (TMR) dextran (red) was used, followed by a second, independent electroporation employing a fresh pipette filled with fluorescein-dextran (green). Notably, $81.6 \pm 1.2\%$ ($n = 3$ mice) of green cells were also red, with almost complete overlap for large projection neurons in deeper layers and a similar recovery rate in superficial layers (Fig. 3f, g). Similarly, using a transgenic mouse line bearing a subset of projection neurons that express a GFP-derived ratiometric indicator protein (Thy1-CLM[19]), NEM electroporation labeled $87.5 \pm 12.5\%$ ($n = 2$ mice) of CLM-expressing projection neurons that are associated with a targeted glomerulus (Supplementary Fig. 3). This indicates that NEM electroporation indeed labels the glomerular module very effectively.

### In-depth circuit architecture of the MOR174–9 glomerulus.
To assess the quantitative capability of this circuit delineation strategy, we used NEM electroporation within a specific, genetically labeled glomerulus (MOR174–9-GFP glomerulus[20,21]) as example of an extended neuronal circuit (Fig. 4a, $n = 5$).

In total, 162 (median; q1 −26.75; q3 +25.75) cells were labeled per glomerulus (Fig. 4b). Approximately 80% of all somata were

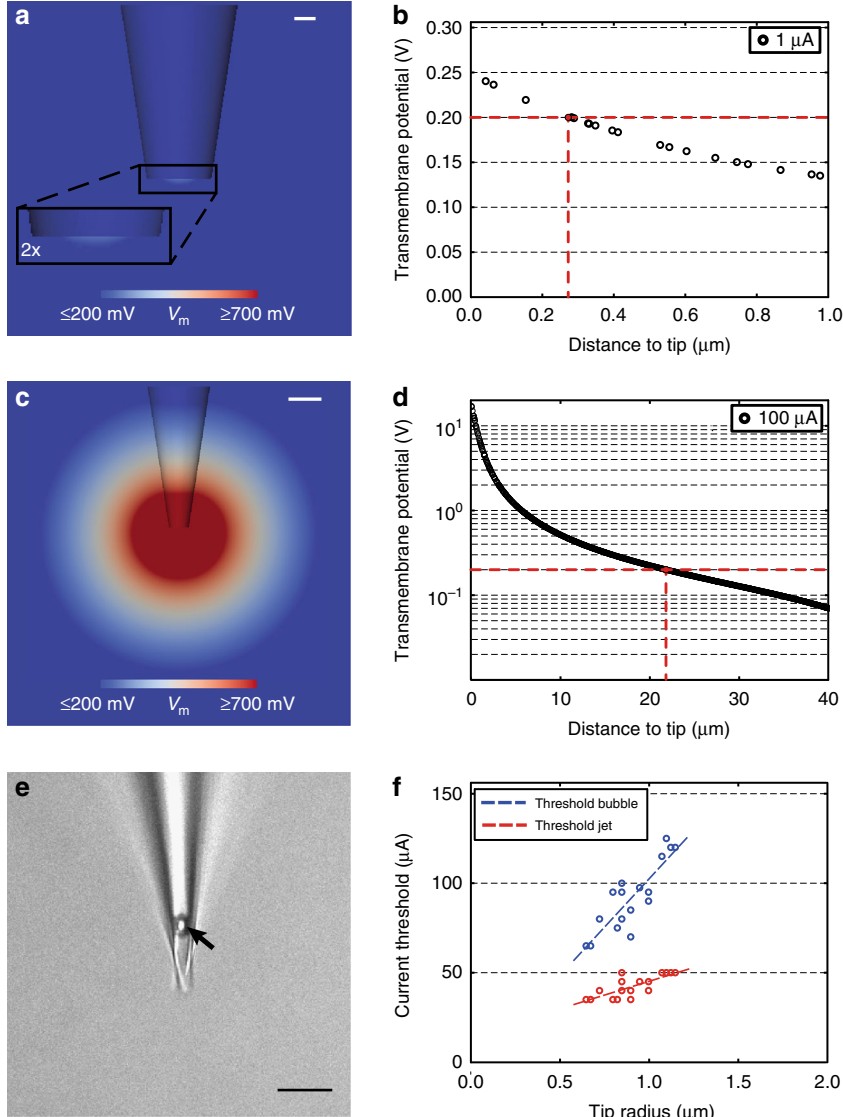

**Fig. 1** Effectiveness of standard glass microelectrode electroporation can be predicted by FEM but is restricted in practice by physical limitations. **a** 3D-FEM model showing the center cut of a standard glass micropipette. The figure illustrates the volume where effective electroporation (transmembrane potential >200 mV) can occur at 1 μA ($V_m$ = transmembrane potential). Inset depicting a twofold magnification of the effective electroporation zone close to the pipette tip. Scale bar = 1 μm. **b** Plot of corresponding voltage drop along the first micrometer of the central axis of the pipette at 1 μA. Black rings indicate individual elements along the central axis of the pipette (assumed electroporation threshold of 200 mV marked by the dashed red horizontal line; resulting critical distance of 0.285 μm indicated by the dashed red vertical line). **c** Center cut of the 3D-FEM model employing a standard glass micropipette at 100 μA, illustrating the volume where effective electroporation (transmembrane potential >200 mV) can occur. Scale bar = 5 μm. **d** Corresponding voltage drop along the first 40 μm of the central axis of the pipette at 100 μA. Black rings indicate individual elements along the central axis of the pipette. Electroporation threshold and critical distance (~22 μm) as in **b**. **e** When increasing stimulus intensities beyond 30–40 μA, a jet-like convection movement and gas bubble (black arrow) formation appear, as seen here in an exemplary camera frame under the x20 objective. Scale bar = 20 μm. **f** Current threshold values (μA) for the jet (red) and gas bubble (blue) phenomenon plotted against tip radius (μm). Dashed lines indicating a linear fit for both ($R^2$ = 0.58 for jet, red and $R^2$ = 0.74 for bubble, blue)

located within 100 μm of the glomerular center. However, the radius to the glomerular center was doubled (from ~180 μm to ~390 μm) in order to include the final 10% of cells, with the most distant soma being located 471.5 μm from the glomerular center (Fig. 4c, d). Since the functional roles of cells or cell types are related to their spatial positions in the gross anatomical organization of the bulb, in a second step all cells were separated into four groups according to their respective layer identities (Fig. 4e–g): the glomerular layer ("GL," $n = 124.2 \pm 39.4$ per glomerulus; 79% of all cells), outer external plexiform layer ("outer EPL," $n = 21.6 \pm 9.5$, 14%), inner external plexiform layer

("inner EPL," $n = 3.2 \pm 0.8$, 2%) and mitral cell layer ("MCL," $n = 9 \pm 1.6$, 6%).

In the architecture of the bulb, the MCL is of particular interest because it is thought to represent the main output element of the bulb, comprising mitral cells (MCs) and deep tufted cells (dTCs). This crucial structure was found to comprise only 6% of all cells constituting the glomerular circuitry. Since NEM labeling is near complete, we can now ask whether the distribution of MCs and dTCs is stereotypical between animals. MCL projection neuron (MCLPN) soma positions were transformed and analyzed onto a common coordinate system (see "Methods" section). Indeed,

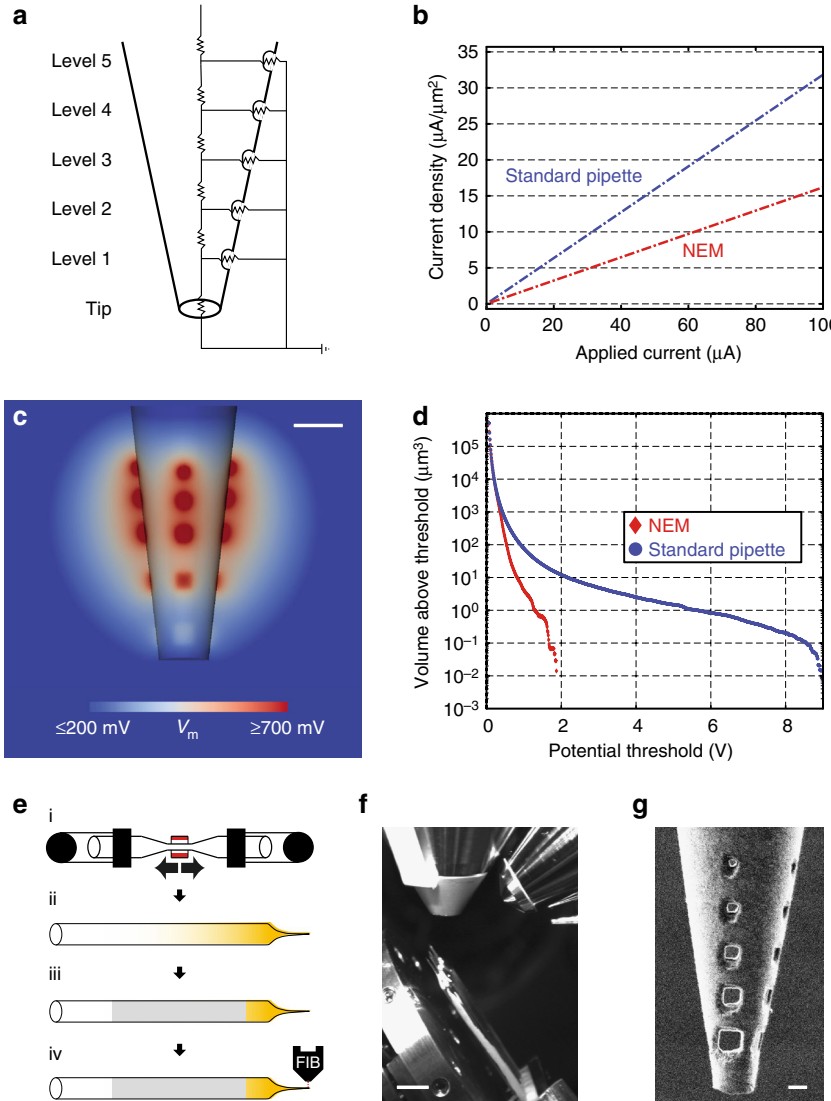

**Fig. 2** Nanoengineered electroporation microelectrodes (NEMs) allow for improved current distribution and electroporation effectiveness by reducing peak potential regions. **a** Scheme showing the current divider-based electric circuit model with five divisions ("levels") along a central axis, serving as an equivalent electric circuit model of the newly designed pipette. For simplicity of the scheme, exterior resistances are neglected. **b** Maximum current density ($\mu A/\mu m^2$) of a standard pipette (blue) and an NEM (red) plotted against applied stimulation current ($\mu A$). The graph shows that current density of the standard pipette rises twice as quickly when compared to the NEM. **c** Cross-section of the 3D-FEM model illustrating the total effective electroporation volume and its distribution around the pipette tip at 50 $\mu A$ employing the NEM ($V_m$ = transmembrane potential). Scale bar = 5 $\mu m$. **d** Cumulated volume of elements for a standard glass pipette (blue) and an NEM (red) beyond a given potential at 50 $\mu A$ (X-axis, (V) in steps of 0.01 V). **e** After pulling long-tapered, patch clamp-like pipettes (i), pipettes were successively coated with a thin gold layer (ii) and conductive silver paint (iii). This was necessary to provide efficient grounding of the surface for FIB-assisted milling (iv). **f** Inside camera view of the vacuum chamber for SEM imaging and FIB-assisted milling. Glass pipettes are mounted and aligned at an angle of 90° relative to the principal axis of the FIB. Gallium ion gun shown in the right upper corner, electron beam mounted vertically in the central portion of the chamber. Scale bar = 5 mm. **g** Example of an NEM after successful insertion of the five-level hole design, as seen in high-resolution FIB imaging mode. Scale bar = 2 $\mu m$

MCLPNs from all experiments ($n = 5$) were clustered within 111 $\mu m$ ($\pm 68.6\ \mu m$) around a common center of gravity. The center position of the cluster was well preserved between animals and was shifted 92 $\pm$ 99.5 $\mu m$ dorsally and 74 $\pm$ 81.8 $\mu m$ posteriorly from the projected position of the glomerulus (Fig. 5a, b).

Recent work has determined that MCs and dTCs exhibit distinct electrophysiological properties in vivo and may thus have different functional roles[22–24]. To distinguish between these two cell types, we fitted a two-component Gaussian mixture model (GMM) to analyzed our morphological data (Fig. 5c-e). In general, there was clear separation between the two cell types (Supplementary Fig. 4a, b). Surprisingly, however, some

cells were found in the MC cluster which had substantially smaller soma areas ($<200\ \mu m^2$) compared to "typical" MCs. These small MCs (sMCs) showed several other morphological differences from typical MCs (Table 1), and likely represented a separate group of rare MCLPNs (see also Supplementary Fig. 4c), and are not further considered here. Therefore, in total, 5.2 ($\pm 1.1$) MCLPNs could clearly be classified as classical MCs and 3.2 ($\pm 1.3$) as dTCs per *MOR174–9* glomerulus (Fig. 5f). While individual MC and dTC midpoint positions had a close spatial relationship to each other (Supplementary Fig. 5), the distribution patterns of MCs and dTCs showed a notable difference between the two subsets, with a substantially larger

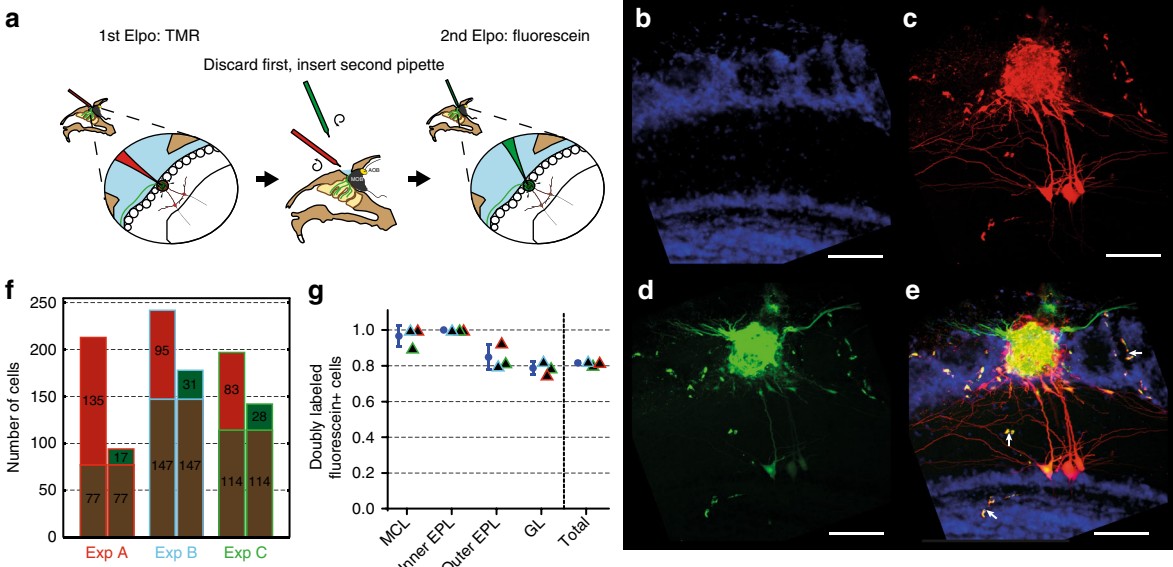

**Fig. 3** In vivo electroporation of glomerular neurons using NEMs. **a** Experimental strategy: In order to probe reliability and comprehensiveness of the method, two differently colored dyes were sequentially and independently electroporated into the same glomerulus. First, electroporation with TMR-dextran. Retraction of first electrode and insertion of the second pipette, after an interval of ~15 min. Second electroporation with fluorescein-dextran. The image panel shows a typical result in the region of interest, as seen by CLSM. **b** DAPI staining revealing the layered structure of the region of interest. **c** TMR dye fluorescence. **d** Fluorescein dye fluorescence, note GFP-marked olfactory sensory neuron axons innervating the *MOR174–9* glomerulus. **e** Overlay of the three channels, note parallel staining of vascular structures (white arrows). Scale bars = 100 μm. **f** Individual data plot of double-labeling experiments (Expts. A–C, n = 3). For each experiment, the left bar shows the number of TMR-positive cells and the right bar the number of fluorescein-positive cells. Each bar is subdivided into doubly labeled fraction of cells (brown) and singly labeled fraction (red for TMR and green for fluorescein). In **g** the fraction of doubly labeled, fluorescein-positive cells is shown for different layers. Blue bar and circle indicate mean ± s.d

spread of MCs along the dorsoventral axis than dTCs (107.4 μm vs. 66.2 μm, Fig. 5g–i).

We further used the data set of glomerulus-affiliated neurons to provide a comprehensive outline of the tufted cell (TC) domain. We define a TC as every neuron that is not an MC that has a primary apical dendrite targeting the glomerulus and also has lateral dendrites. This definition does not comprise the subpopulation of external tufted cells that typically do not possess lateral dendrites. First, a depth profile of the presumed TCs was determined with regard to relative soma position along the deep-to-superficial axis (Fig. 6a). Within the EPL, a sharp increase in the number of TCs was noted from the inner three quarters to the superficial quarter. Therefore, TCs from these regions were separated from each other and were named "middle TCs" (mTCs) and "superficial TCs" (sTCs), respectively. On average, only 3.2 (±0.8) mTCs per glomerulus were found. On the other hand, 18.8 (±4.4) sTCs per glomerulus were found, thereby constituting 90% of glomerulus-associated cells from the superficial quarter of the EPL that meet our criteria for TCs. Neurons from both subsets shared distinct common morphologic features (Table 1).

To investigate the exact spatial relationship of these domains along the deep-to-superficial axis, dendrite distributions were compared among the four cell types (Fig. 6b–f and Supplementary Fig. 6). In all cases considered, a highly significant separation was found ($p < 0.001$ using a one-sample $t$ test). On average, MC and dTC dendrites were separated by 34.27 (±33.96) μm, dTC and mTC by 25.78 (±42.21) μm, and mTC and sTC by 44.83 (±39.18) μm. The corresponding spatial "overlap," i.e., difference between two subsets in the non-dominant direction, was 16% between MCs and dTCs, 21% between dTCs and mTCs, and 13% between mTCs and sTCs. These results indicate that soma position precisely predicts the average dendritic domain within the EPL, which is reflected by the relative dendritic contributions from the four cell types per EPL subregion (Fig. 6g).

**TC axon collateral hotspots associate with MCLPN cluster.** Detailed anatomy, including the ability to stain individual axons, allowed us to examine the local axonal projection pattern of the projection neurons more closely. While TCs are known to give rise to extensive local axonal collaterals in the internal plexiform layer (IPL)[25,26] MCs do not[23], and their axon collaterals course more distantly in the granule cell layer (GCL). Thus, stained axonal processes in the local IPL were considered putative TC collaterals (pTC). In individual experiments, an association between projection neurons and pTC axon collaterals in the adjacent IPL could be observed, and reconstructed axons from more superficially located TCs were found to target the IPL region adjacent to identified MCs and dTCs (Fig. 7a, b). To analyze this apparent association quantitatively, we determined pTC axonal density in projection maps of the local IPL ($n = 3$) (detailed in "Methods" section). Irrespective of exact cellular identity, matching MCLPNs indeed significantly colocalized with pTC collaterals, compared to a shuffled control (see "Methods" section; two-sample $t$ test: for MCs $p < 0.001$ for dTCs $p < 0.005$, for both subsets together $p < 1 \times 10^{-6}$). These results provide evidence for a previously unexpected high association of MCLPNs and pTC axon collaterals from the same glomerulus (Fig. 7c–g). Volume electron microscopy using serial block-face scanning electron microscopy[27–29] allowed us to further follow pTC axons and identify synapses onto GCs in the IPL (Supplementary Figs. 7 and 8), whereas GCs in turn can indeed make synapses onto somata of MCs directly above them.

Thus, NEM electroporation permits quantification of total cell number, the spatial extent of projections into an olfactory glomerulus, the stereotypy of soma locations and a comprehensive analysis of most neurons directly projecting into a glomerulus. Furthermore, this near-completeness allows detection of higher-order features, such as the correspondence of pTC

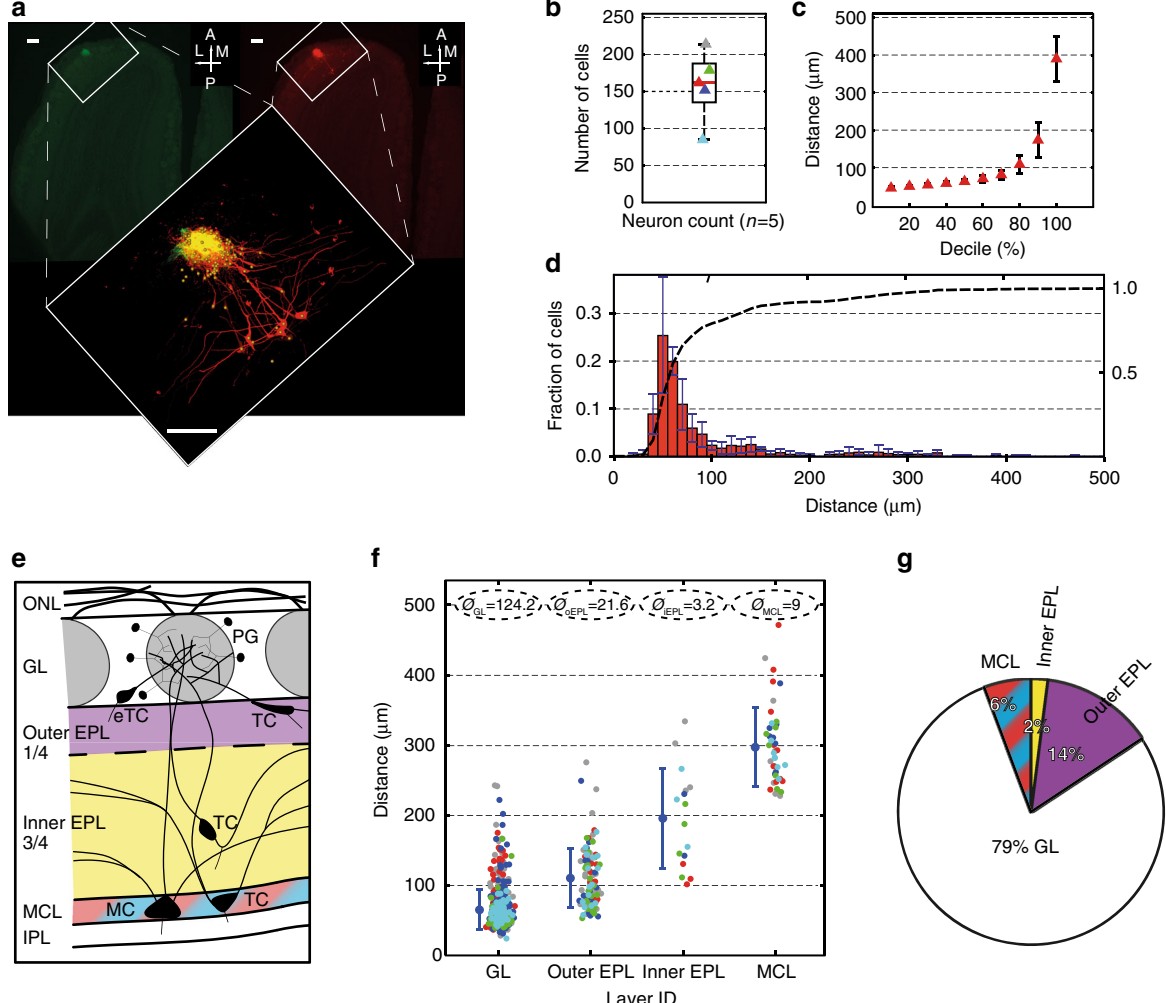

**Fig. 4** Gross cellular analysis of the *MOR174–9* domain. **a** Typical result of the experimental approach. Left image showing the GFP-positive *MOR174–9* glomerulus in a horizontal slice of the olfactory bulb. Right image showing TMR-dextran fluorescence after successful electroporation. Scale bars = 100 μm. Blow-up: overlay of the two channels with marked cell bodies (yellow). Scale bar = 100 μm. **b** In total, a median of 162 (Q1 +25.75; Q3 −26.75) cells (*n* = 5 animals) projecting to the electroporated *MOR174–9* glomerulus is found. Colored triangles indicate the total number of cells for each individual experiment. **c** Variability of percentile borders (i.e., distance to the center ± s.d. (μm)) across all experiments as a function of decile. **d** Relative Euclidean distance distribution of the cells associated with the "standard" *MOR174–9* glomerulus (red vertical bars, bin size of 10 μm). Red bars refer to the left *Y*-axis and show the fraction of cells within each bin and the corresponding SDs among the five experiments (dark blue bars). Dashed black line represents the cumulative plot of the histogram and refers to the right *Y*-axis. **e** Illustration indicating the four layers of the bulb, which are considered separately: GL (white), outer quarter of the EPL (magenta), inner three quarters of the EPL (yellow), and MCL (red-cyan). For completeness, olfactory nerve layer (ONL), IPL, and main cell types are also shown. PG = periglomerular cell, eTC = external tufted cell, TC = tufted cell, MC = Mitral cell. **f** Scatter plot of the distances of all cells (*Y*-axis) to the glomerular center separated by layer identity of the cell soma localization (*X*-axis). Same color code of the dots as in **b**. Numbers in the dashed bubbles indicate mean number (∅) of cells of each layer per glomerulus. **g** Pie chart illustrating the average relative cellular composition of the *MOR174–9* glomerulus, according to layer identity

axons and MC soma locations, consistent with a parallel inhibitory feedforward pathway.

## Discussion

Exhaustive yet specific delineation of neural circuits and projections remains a major challenge with current labeling techniques. Here, we present the development of nanoengineered electroporation microelectrodes (NEMs) as a new tool for predictable, volumetrically exhaustive manipulation of neuronal networks, providing missing input and output functions to the emerging field of dense circuit reconstruction techniques. With increasing availability of nanoengineering devices in engineering and neuroscience departments[30,31], fabrication of customized tools to deliver or record electric fields to/from neural tissue at the nanoscale becomes feasible. Our proposed microelectrode design requires only a few additional steps beyond a simple pulling procedure of glass capillaries, and is thus facile to implement. Certainly, fabrication design is not restricted to this specific approach: redistribution and number of release sites are essentially unlimited and tailored solutions for almost any geometry can be implemented. Even complete nanofabrication with silicon technology including nanostraws[32] has become feasible – with the main challenge remaining how to distribute the electric current.

We have shown that typically-used stimulation intensities for targeted electroporation reliably cover only a limited volumetric range, and our modeling results here are in excellent agreement

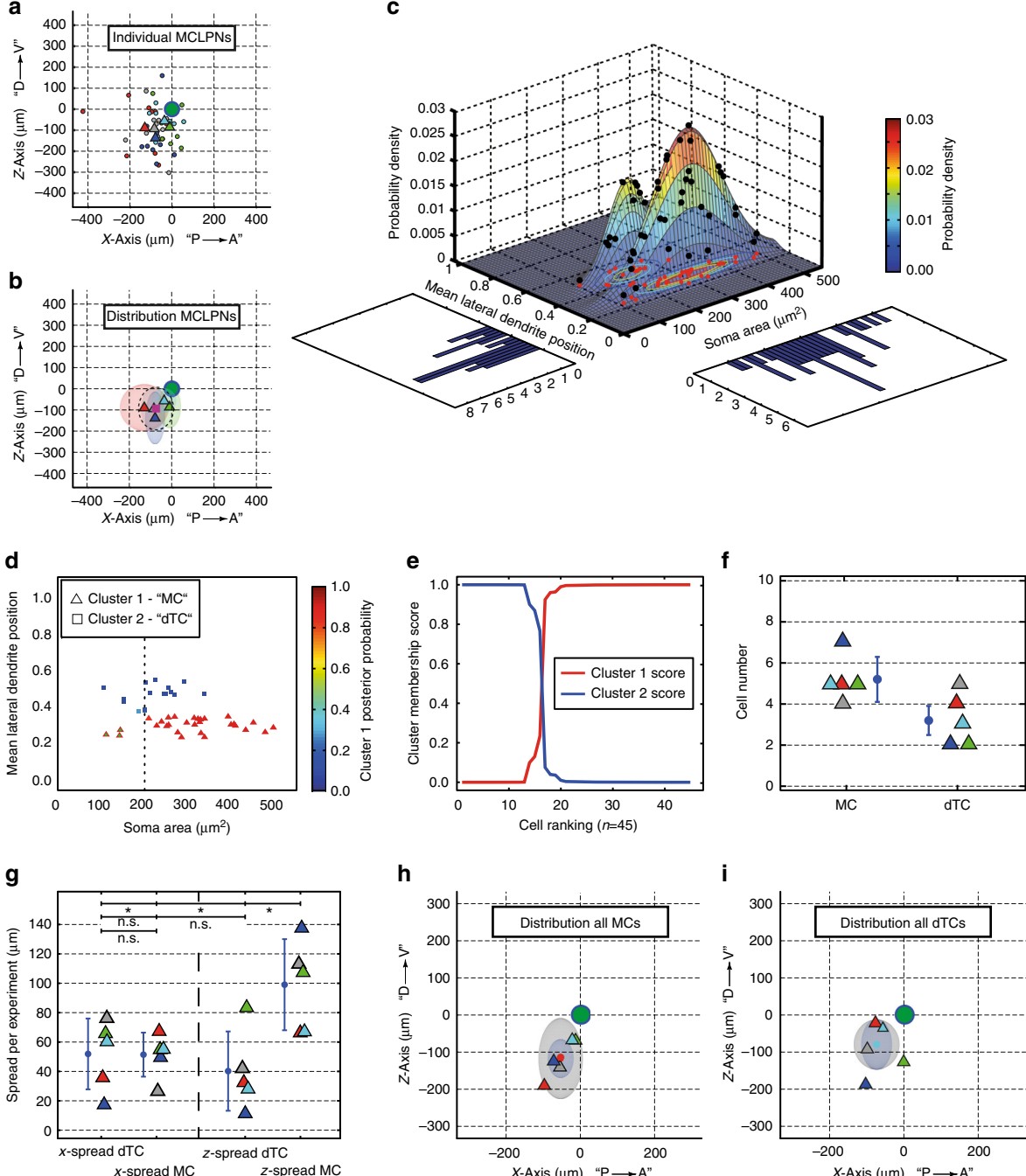

**Fig. 5** Morphological classification of projection neurons in the mitral cell layer. **a** X–z-plane projection of all mitral cell layer projection neurons (MCLPNs, n = 45). X-axis approximately corresponding to posterior (P) → anterior (A) and Z-axis to dorsal (D) → ventral (V) direction. Triangles represent MCLPN midpoints of the five experiments. **b** Overlay of preferential MCLPN locations indicated as ellipsoid body around the respective midpoint based on SD of distances (n = 5). Magenta square indicates common MCLPN midpoint and dashed line denotes SD of the distances of all cells. **c** Based on two principal morphological parameters, MCLPNs were analyzed by fitting a two-component GMM. Probability density function (PDF) shown as discretized mesh (coloring as indicated). Black dots indicate individual cells, red dots their projections onto the 2D plane. Colored PDF isolines according to color bar. Histograms show frequencies of individual dimensions. **d** Consecutive assignment of cells to one of the two clusters (triangles: Cluster 1, MC or squares: Cluster 2, dTC). Coloring indicates posterior probability of each cell to be part of the MC cluster. Dashed line shows cutoff value of 200 μm² soma area, the assumed lower limit for typical MCs. Cells of the MC group having a smaller soma size are individually marked by green dots. **e** Cell ranking according to "cluster membership score," derived from posterior probabilities. **f** Cell numbers of the two subsets MC and dTC (n = 5, mean ± s.d.). **g** Comparison of x- and z-spreads per experiment between MCs and dTCs (two-sample t test, *p < 0.05, n.s. not significant). Mean ± s.d. indicated by blue circle and bar. **h** Ellipsoid body based on SDs of distances to common MC midpoint. The smaller inner ellipsoid (blue) around the common MC midpoint (red circle) is based on SD's of the distances from individual MC midpoints to the common MC midpoint. **i** Same for dTCs. Green circle illustrates the position of the glomerulus at the origin. Color code for data from different experiments as in Fig. 4b

**Table 1 Principal parameters of the projection neurons**

|  | MC | sMC | dTC | mTC | sTC |
|---|---|---|---|---|---|
| Cell number per glomerulus | $5.2 \pm 1.1$ | $0.6 \pm 0.9$ | $3.2 \pm 1.3$ | $3.2 \pm 0.8$ | $18.8 \pm 4.4$ |
| Mean lateral dendrite position in EPL[a] | $0.30 \pm 0.03$ | $0.25 \pm 0.02$ | $0.47 \pm 0.10$ | $0.61 \pm 0.17$ | $0.90 \pm 0.06$ |
| Soma area ($\mu m^2$) | $333 \pm 79$ | $132 \pm 18$ | $221 \pm 58$ | $161 \pm 56$ | $84 \pm 29$ |
| Relative soma position in EPL[b] | 0 | 0 | 0 | $0.48 \pm 0.25$ | $0.93 \pm 0.06$ |
| Number of lateral dendrites | $3.6 \pm 0.8$ | $1.7 \pm 0.6$ | $2.8 \pm 1.0$ | $2.3 \pm 0.7$ | $1.8 \pm 0.8$ |
| Number of secondary dendrites along apical dendrite | $1.4 \pm 0.7$ | $1.3 \pm 0.6$ | $1.3 \pm 0.5$ | $0.6 \pm 1.0$ | N/A |
| Most distal apical dendrite bifurcation in EPL | $0.21 \pm 0.11$ | $0.13 \pm 0.01$ | $0.40 \pm 0.15$ | N/A | N/A |
| Most superficial lateral dendrite extension in EPL | $0.53 \pm 0.10$ | $0.77 \pm 0.21$ | $0.72 \pm 0.06$ | $0.79 \pm 0.12$ | N/A |
| Most internal lateral dendrite extension in EPL | N/A | N/A | N/A | $0.31 \pm 0.23$ | N/A |
| Relative position of axonal origin in EPL | N/A | N/A | N/A | $0.33 \pm 0.24$ | N/A |
| Distance from axonal origin to soma ($\mu m$) | N/A | N/A | N/A | $45 \pm 24$ | N/A |

[a] These two parameters served as the basis for cluster analysis of MCL cells (comprising MC, sMC, and dTC)
[b] This parameter was taken to separate mTC from sTC at a cutoff value of 0.75; by definition, all MCL cells (MC, sMC, and dTC) have a value of 0

with our own (Supplementary Fig. 1) and published[10,14] data. To access higher stimulation intensities and thus larger volumes, however, it was necessary to reduce peak potentials because these led to mechanically harmful jet and gas bubble formation in conventional pipettes, and greatly surpassed the assumed hazardous transmembrane potential of 700 mV known to cause lysis for a number of cell types[18]. The achieved cumulated volume >700 mV corresponds to a conventional pipette at approximately 20 μA stimulation intensity, which is still larger than values that were tested safe for physiological network function[11]. This may be the reason why we found fewer labeled cells following a second electroporation epoch in our double electroporation experiments, indicating some residual damage especially among the population of smaller neurons[33,34]. For probing physiological functions, NEM design should therefore further aim to reduce peak potentials by inserting even more release sites, or implementing an array design that includes several NEMs.

The key target of our approach was, however, completeness of cell loading, which can only be assessed indirectly in the absence of a "ground truth". We first used overlap of two independently labeled subsets of cells in the same glomerulus as a reliability measure and simplified surrogate parameter of completeness. Among all cells labeled, we found an overlap of more than 80 % while overlap among neurons residing in the inner layers of the MCL or EPL was close to 100 %, underlining a near-complete delineation of the circuit, especially of the MC and TC domains. Additionally, the determined total number of neurons corresponds well to average global estimates in the literature when adjusting for the most recent number of glomeruli in mice which was found to be far higher than previously assumed[35–37] (see also Supplementary Discussion). However, it must be acknowledged that the volumetric range of our proposed stimulation settings lies below the whole volume of the considered target circuit (40–50 μm radius), in order not to compromise the required specificity when handling the pipette at the microscale. Very small cellular processes passing only peripherally through the glomerulus might therefore still be missed by electroporation. Indeed, while relatively straight, unbranched passing processes resembling "juxtaglomerular association neurons"[38,39], a cell type which is believed to span multiple glomeruli, could be found in neighboring glomeruli, we did not identify labeled somata at distances >300 μm from the electroporation site within the superficial layers (Supplementary Fig. 9a). This may hint at the possibility that axonal structures are not as efficiently loaded as dendrites by the proposed protocol, and that a dedicated strategy might have to be used for such structures, potentially involving the adjustment of the dextran molecular weight or the use of a different marker. Nevertheless, our technique is generally capable

of providing retrograde axonal filling, because olfactory nerve fascicles (which contain olfactory sensory nerve axons) can indeed be electroporated successfully and their corresponding glomerulus labeled (Supplementary Fig. 9b). The need for volumetrically exhaustive labeling approaches becomes particularly evident when sparse, higher-order features of a neuronal circuit, which have so far remained elusive to statistical or local labeling approaches, are under investigation. Using NEM electroporation we provide anatomical evidence for an earlier suggested cluster of main projection neurons in the MCL[40,41] whose size and location was found to lie in a range similar to what has been proposed by standard electroporation[12,13,20,42]. Interestingly, the position of this cluster relative to the position of the glomerulus is well preserved between animals, and the positional variability is substantially lower than what has been reported for the variability of a glomerulus itself[43,44]. Within the cluster, MCs show almost twice the spread in the dorsoventral direction compared to either the anteroposterior direction or to dTCs in any direction. This suggests a wider integration of bulbar olfactory information in the MC subunit given the arrangement of olfactory receptor domains in the bulb[16]. It furthermore suggests that MCs might receive perisomatic inhibition from more broadly distributed GCs. Among glomerulus-associated MCLPNs, we could identify a small MC cell type in 40% of glomeruli, which was morphologically clearly distinct from "typical" MCs or dTCs. However, this neuron type has been largely neglected in the literature, and we only found a similar early drawing by Ramón y Cajal[45]. Further work will be needed to establish its functional role in the context of glomerular circuitry.

We found an exquisite laminar stratification of lateral dendrites from different types of projection neurons, matching earlier reports on the sublayers of the EPL using sparse labeling[26,46,47]. However, our strictly quantitative examination showed only minimal degrees of overlap between these types of cells, with lateral dendrite position within the EPL representing a categorical property rather than a mere preference. Interestingly, although only representing a small group of cells, even mTCs appeared to occupy a distinct subband, with spatial location predicting dendritic position. Therefore, the local information flow of the main bulbar cell types in the EPL appears to be handled in a spatially segregated, "floor-like" fashion, with very little direct dendrodendritic information exchange between lateral dendrites of different cell types to be expected. If such interconnections exist, they must be accomplished by an additional "vertical" processing unit like distinct GCs[47,48] or short axon cells[49].

While intrabulbar axon collaterals from TCs have been known to underlie the interplay of the two mirror-symmetric glomeruli for more than a decade[50–52], the target and function of local TC

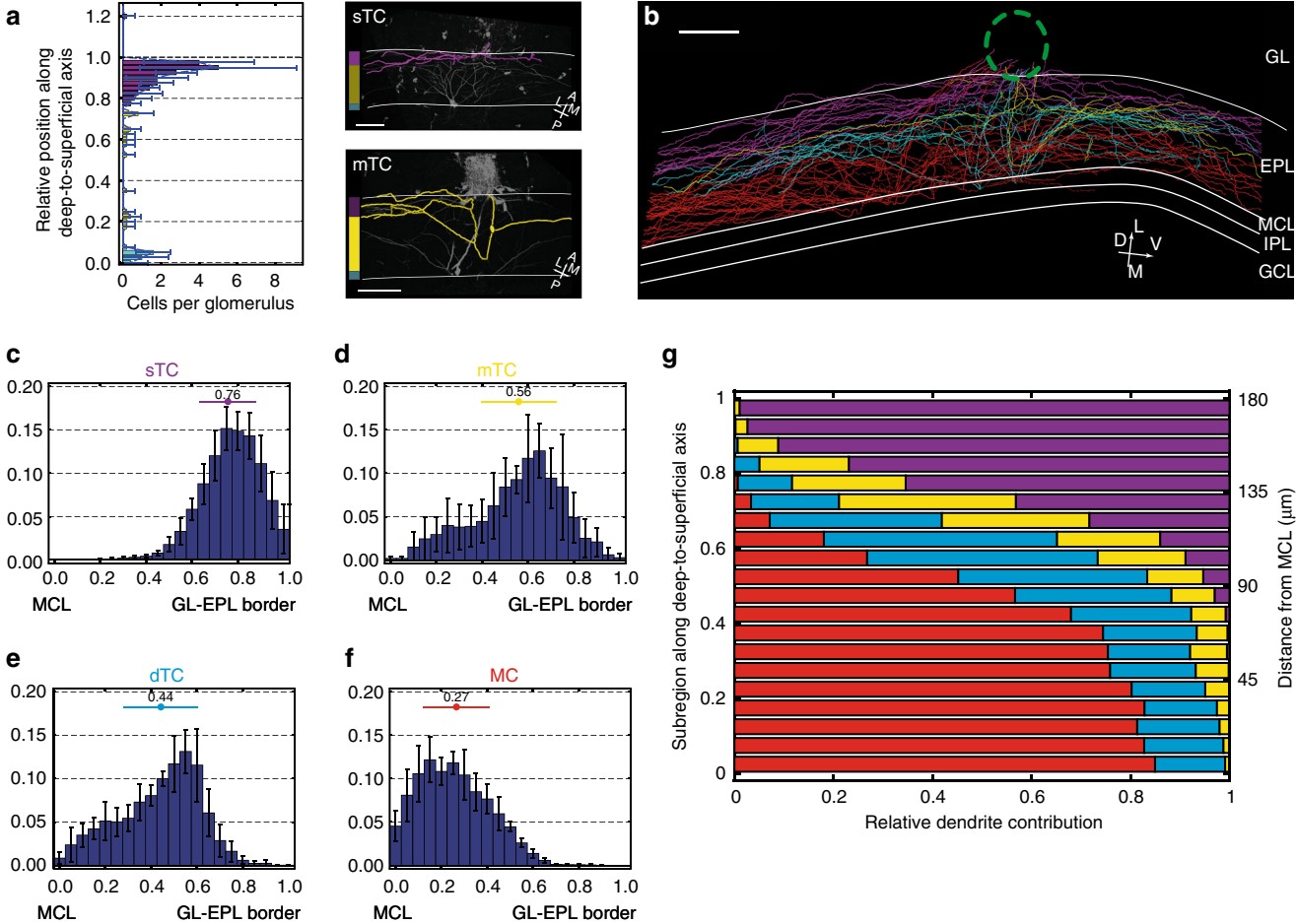

**Fig. 6** Different types of projection neurons form segregated dendritic bands within the EPL. **a** In the horizontal histogram on the left, average number of neurons per glomerulus of the three separately analyzed TC subregions plotted along the deep-to-superficial axis from the inner edge of the MCL (0) to the GL-EPL-border (1) in bins of 0.025. Bar color represents subgroup of TC (magenta = sTC, yellow = mTC, cyan = dTC). Blue bars indicate ± s.d., dashed blue line adds a line representation. Right, examples of sTCs and mTCs in horizontal partial volume sections of a CLSM image stack. Outline of cellular branching pattern of these cells colored in magenta and yellow. White lines indicate GL-EPL border (top) and MCL (bottom), respectively. Scale bars = 100 µm. Orientation as indicated (A = anterior, P = posterior, L = lateral, M = medial). **b** Exemplary whole dendritic reconstruction of all labeled *MOR174–9* projection neurons of one experiment. The local dendritic projectome shows a strict laminar pattern when neurons are separated by cellular identity (magenta = sTC, yellow = mTC, cyan = dTC, and red = MC dendrites). Dashed green circle represents the electroporated glomerulus. IPL = internal plexiform layer, GCL = granule cell layer. Scale bar = 100 µm. Orientation as indicated (V = ventral, D = dorsal, L = lateral, M = medial). **c–f** Histogram plots showing the average distribution of lateral dendrites within the EPL from the four cell types, in bins of 0.05 from the MCL (0) to the GL-EPL border (1) (**c** = sTC, **d** = mTC, **e** = dTC and **f** = MC). Black vertical bars indicate ± s.d. Overall mean dendritic position for each cell type is shown ± s.d. (number and colored point and horizontal bar above each histogram). **g** Average relative dendrite contribution from the four cell types (magenta = sTC, yellow = mTC, cyan = dTC, red = MC) per subregion of the normalized EPL width in bins of 0.05. Right *Y*-axis showing the contributions as absolute distance from MCL within a typical EPL width of 180 µm

axon collaterals have remained enigmatic[23,25,26,53]. In our exhaustive labeling study, we found anatomical evidence for an unexpected association of TC axons and MCs sharing the same parent glomerulus, with TC axons projecting specifically to these MC soma locations in the adjacent IPL. Given the reported synapses onto proximal GC apical dendrites from horizontal, presumed axonal branches[54,55] as well as the distinct physiological behavior of MCs and TCs in vivo[22–24], this association suggests a feedforward TC → GC → MC pathway. Consistent with this, volume electron microscopy showed putative TC axons synapsing onto GCs, as well as GC-MC axo-somatic synapses (Supplementary Figs. 7 and 8) although unambiguous direct evidence of such a triplet circuit is still missing. Furthermore, this finding provides a complementary or even alternative explanation for the observation of translaminar olfactory columns as revealed by retrograde-specific viruses[56,57].

Neuroanatomy is undergoing an impressive renaissance. Sparse higher-order features of extended neuronal circuits, however, have so far been elusive because they require volumetrically exhaustive yet specific circuit manipulation techniques. NEM electroporation therefore provides a powerful tool to complement dense circuit reconstruction techniques in the future, by mapping inputs and outputs of a circuit of interest and thus defining the minimal geometric requirements needed for reconstruction. Furthermore, the range of applications may be extended to other molecules, such as loading local neurons with DNA constructs.

## Methods
**Finite element modeling and implementation of FEM**. To simulate the potential distribution of electroporation electrodes, the finite element method was used[58,59]. Laplace's equation $\nabla^2\phi_{ind}=0$ was numerically solved for a 3D space filled with a homogeneous electrolyte using appropriate boundary conditions, where $\nabla^2$ is the

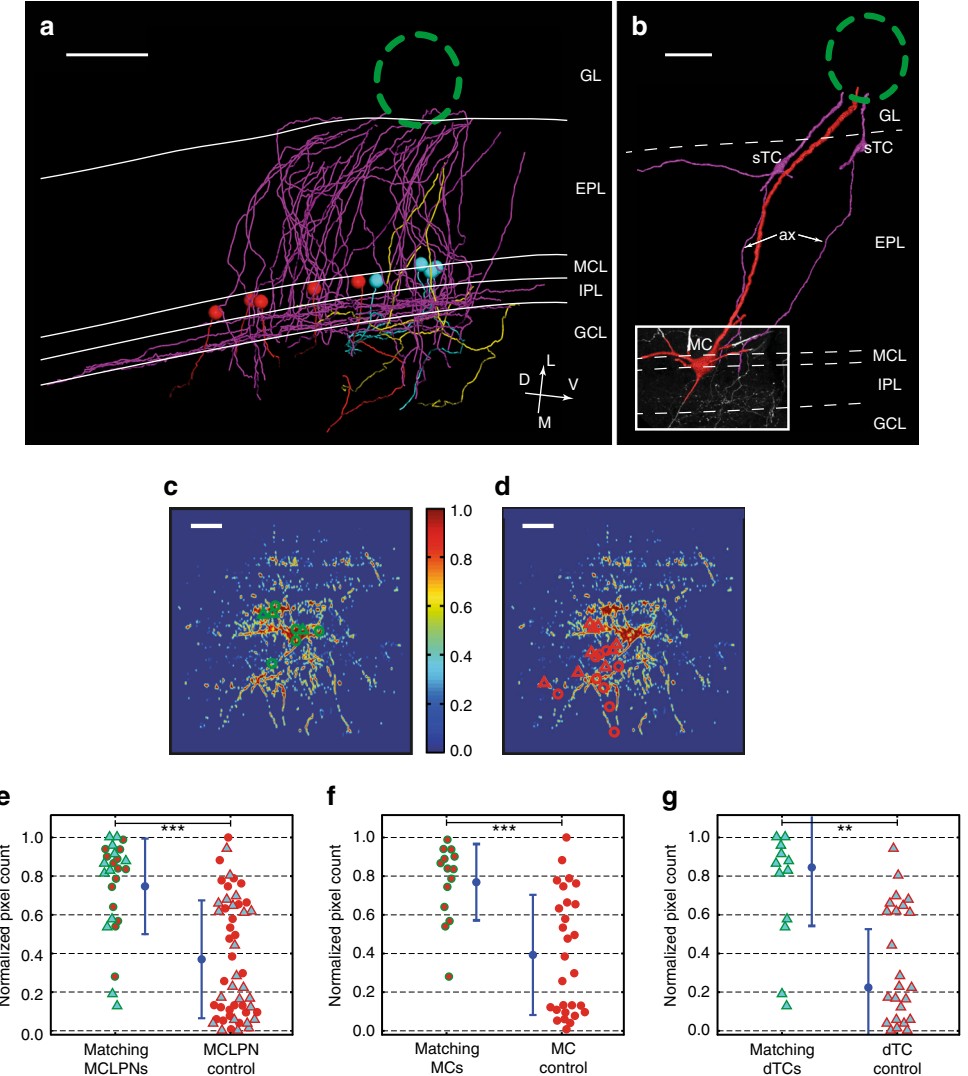

**Fig. 7** Tufted cell axons project specifically to MCLPN soma locations. **a** Axonal projections from superficially located TCs traverse the EPL in a column-like arrangement towards the IPL, next to the cluster of the MCLPNs of the same glomerular domain. The same color code is used as in Fig. 6b, indicating different cell types. Red and cyan spheres show the soma positions of MCs and dTCs, respectively. No recurrent axon collaterals from MCs and only minor contributions from dTCs can be found. By far, most axonal branches arise from sTCs. Scale bar = 100 μm. **b** Example of a 3D-rendered MC (red) and two corresponding sTCs (magenta), which project their axons ("ax") specifically to the IPL adjacent to the position of the MC, where they support a local collateral mesh. Inset box showing a maximum projection view of the local IPL (depth = 60 μm) next to the MC from a CLSM image stack. Scale bar = 50 μm. **c** Local axonal IPL mask with positions of the corresponding MCLPNs of the same experiment shown as an overlay (transparent green circles indicating MCs, transparent green triangles marking dTCs). Color bar representing average pixel values ranging from 0 to 1. **d** Overlay of MCLPN positions form the other two experiments indicated as an intrinsic "shuffled" control (transparent red circles indicating MCs, transparent red triangles marking dTCs). Scale bars = 100 μm. **e** Normalized pixel counts of all MCLPNs of the three experiments ("matching," = green lining, MCs = red filled circles, dTCs = cyan filled triangles) are compared to counts of non-matching MCLPN positions from the other two experiments ("control," = red lining). **f**, **g** Same comparison shown for MCs and dTCs individually. Mean values ± s.d. shown in blue. Two-sample $t$ test, **$p < 0.01$, ***$p < 0.001$

Laplace operator and $\phi_{ind}$ is the induced electric potential. In agreement with our experimentally used physiological saline solution, a specific conductivity of 1.58 S/m was assumed. The compound transmembrane potential $\Delta\phi_{trans}$ was assumed to be made up by an expected natural neuronal resting potential $\Delta\phi_{rest}$ of −70 mV and the induced potential by the electroporation device $\Delta\phi_{ind}$, resulting in $\Delta\phi_{trans} = \Delta\phi_{rest} + \Delta\phi_{ind}$. The geometric design and generation of an unstructured 3D calculation mesh were done using Gmesh[60] software, implementing its built-in geometrical scripting language and the netgen algorithm[61]. Mesh density decreased with increasing radial distance to the pipette axis, and the electric field drop was expected to be highest in close vicinity to the pipette, thus our computational precision is less significant at greater distances. The finite element simulation itself was set up and conducted with ElmerSolver (CSC Finish IT Center for Science, ElmerSolver manual, http://www.csc.fi/english/pages/elmer) using the implemented BiCGStab method to solve the matrix equation, with an ILUT preconditioner to achieve faster convergence (tolerance level $1 \times 10^{-8}$). For post-processing and visualization, the Visualization Toolkit[62] and Paraview[63] were used.

Subsequent volumetric calculations were run in Matlab (Version R2011a, The MathWorks, Natwick, MA). A volume was calculated in which the expected transmembrane potential surpassed an assumed threshold value of 200 mV, the threshold assumed for efficient electroporation[17]. Furthermore, the volume exceeding an upper limit of 700 mV was determined, assuming this to be the threshold of cell lysis[18].

**Physical model and boundary conditions.** The pipette was modeled as a truncated cone. In agreement with scanning electron microscopy (SEM) measurements of a typical electrode (Fig. 2g and Supplementary Fig. 10), the outer diameter of the pipette at 25 μm proximal to the tip was measured to be 10.6 μm, and the tip radius was 2 μm for the NEM and 1 μm for of the assumed "standard" electroporation pipette[11,14,64]. Glass thickness was 0.5 μm. The pipette tip was centered in an 80 × 80 × 80 μm bounding box, under the assumption that the bulk of the potential drop occurred within these limits. The current density $J$ on the surface of the pipette was

assumed to be 0, except at the tip and the inserted holes where it was set according to the calculated released currents $I$. These were obtained by a pen and paper calculation, modeling the holes and the tip as a current divider circuit. Inside the holes, current density was assumed to be uniform so that the relation $I = JA$ holds, with $A$ being the surface.

**Fabrication of microelectrodes.** For standard patch-like microelectrodes, thick-walled borosilicate glass (O.D. 2.0 mm, I.D. 1.0 mm, Hilgenberg, Malsfeld, Germany) was used and pulled to a tip size of 1–2 µm in diameter on a Flaming/Brown type P-97 micropipette puller (Sutter Instruments, Novato, CA, USA). For fabrication of NEM, standard thin-walled borosilicate glass (O.D. 2.0 mm, I.D. 1.7 mm, Science Products GmbH, Hofheim, Germany) was used. Final tip size was 3.5 to 4.5 µm in diameter. As the ability of the internal electrolyte to serve as a current divider critically depends on actual tip geometry (Supplementary Fig. 11), we chose a shallow tip cone with a tip angle $\theta$ of 22°. Next, glass pipettes were placed in a high vacuum sputtering and coating system (Bal-tec MED 020, Leica, Wetzlar, Germany) and heat-coated twice for 40 s with a piece of gold wire (diameter 0.2 mm, Plano GmbH, Wetzlar, Germany). Subsequently, pipettes were painted with a conductive silver paint (SPI, Structure Probe Inc., West Chester, PA, USA) to prevent pipettes from overcharging during application of FIB. Coated and painted glass microelectrodes were then placed in a combined FIB-SEM workstation (Neon 40EsB, Zeiss, Oberkochen, Germany), equipped with a gallium ion source for FIB-assisted milling. By using the gallium ion gun at an angle of 90° to the principal axis of the pipette, square-shaped holes were milled into the glass wall along the shaft, up to an axial distance of 25 µm from the tip, each set apart by 5 µm. The edge length ($s$) of individual holes decreased with increasing distance to the tip ($s_1 = 2.12$ µm, $s_2 = 1.65$ µm, $s_3 = 1.25$ µm, $s_4 = 0.82$ µm and $s_5 = 0.6$ µm). Milling current and dwell time per area were chosen such that every shot could penetrate through both glass walls. Next, pipettes were rotated axially by 90 degrees and the procedure was repeated. Finally, the cone assumed a 5-level, cross-like pattern having the largest openings close to the tip.

**Electric potential measurements in vitro.** For in vitro electric potential measurements, a recording and a stimulation electrode, both standard patch-like glass microelectrodes, were filled with extracellular Ringer's solution (NaCl 135 mM, KCl 5.4 mM, HEPES 5.0 mM, MgCl$_2$ 1 mM, CaCl$_2$ 2.0 mM, pH 7.2, 280 mOsm/kg) and immersed in a bath containing the same Ringer's solution at room temperature. Electrodes were placed in the same focal plane at defined distances to each other and monitored under a standard upright microscope (Zeiss, Wetzlar, Germany). The grounding electrode was placed near the edge of the bath. Both stimulating and recording electrode as well as the ground were composed of chlorided silver wire. At each distance, trains of stimulation pulses of defined intensities (10–50 µA) were delivered under a typical electroporation regime (25 ms pulse length delivered at 2 Hz) and recorded by a microelectrode amplifier device (Axoclamp-2B, Molecular Devices, Sunnyvale, CA, USA). Data were digitized by a converter board (ITC-18, AutoMate Scientific Inc., Berkeley, CA, USA) and read out by the software IgorPro (WaveMetrics Inc., Portland, Oregon, USA) on a standard desktop computer.

To test formation of gas bubbles, only the stimulation electrode was monitored and filmed at x40 magnification while steadily increasing current intensity per pulse (5 µA per second), while using typical electroporation settings (25 ms pulse length delivered at 2 Hz).

**In vivo surgical procedures.** All experimental procedures were performed according to the animal welfare guidelines of the Max Planck Society and the guidelines of German animal welfare law. For two-photon targeted electroporation of individual glomeruli, a genetically modified mouse strain was used that expresses GFP under the promoter of the *mouse olfactory receptor (MOR) 174–9*[20,21]. Experiments targeted the dorsal *MOR174–9* glomerulus. As reported for other glomeruli[43,44] its position was variable (between 10% and 22% on the anterior-posterior axis and 51–62% on the medial-lateral axis, $n = 10$ bulbs), so targeting was based on GFP fluorescence. For confirmatory experiments the method a transgenic mouse line (*Thy1-CLM*) was also used, which is known to exhibit Thy1-driven expression of the ratiometric indicator protein CLM in a subset of projection neurons of the olfactory bulb[19]. *MOR174–9* or *Thy1-CLM* animals (P35–42) were anaesthetized by intraperitoneal injection of meditomidin 0.75 mg/kg, fentanyl 0.025 mg/kg and midazolam 10.00 mg/kg. Body temperature was kept at 37 °C throughout the procedure. After removing the scalp and exposing the cranial bone, the periosteum was thoroughly removed using a scalpel. Next, a metal head plate was attached to the skull using instant adhesive Loctite 4011 (Loctite Corporation, Rocky Hill, CT, Mississauga, Ontario). After clean exposure of the area above the anterior part of the right MOB, a sealing well was formed of Paladur dental acrylic (Heraeus Holding GmbH, Hanau, Germany). Additionally, an AgCl grounding electrode was attached to the well. The skull overlaying the anterior part of the right MOB was thinned using a dental drill. To prevent breaking or clogging of the pipette, the dura mater was removed. Thirty–sixty min after completion of the electroporation protocol, animals were sacrificed by transcardial paraformaldehyde perfusion (4% with pH adjusted to 8.9).

**In vivo imaging and electroporation.** Mice were placed in a custom built two-photon microscope[65] equipped with a Ti-Sapphire laser (Coherent Inc., Santa Clara, CA, USA) and the sealed well filled with extracellular Ringer's solution (NaCl 135.0 mM, KCl 5.4 mM, HEPES 5.0 mM, MgCl$_2$ 1.0 mM, CaCl$_2$ 2.0 mM, pH 7.2, 280 mOsm/kg). To focus on the *MOR174–9-GFP* glomerulus, a ×16 water immersion objective (0.8 NA, Nikon) was used. For two-photon excitation, the laser was tuned to 880 nm. NEMs were tip-filled with extracellular Ringer's solution (NaCl 135 mM, KCl 5.4 mM, HEPES 5.0 mM, MgCl$_2$ 1 mM, CaCl$_2$ 2.0 mM, pH 7.2) containing either tetramethylrhodamine (TMR) (8.3 mg/ml) or fluorescein (12.5 mg/ml) 3000 MW dextran-conjugated dyes (Invitrogen, Life Technologies GmbH). Complementary backfilling was performed with plain Ringer's, as described above. Pipettes were mounted on a micromanipulator (Luigs & Neumann, Ratingen, Germany) and the grounding electrode of the well connected to the ground of a stimulus isolation unit (ISO-STIM 01D, npi electronic GmbH, Tamm, Germany).

Electroporation settings were 25 ms, 50 µA strong square pulses delivered at a frequency of 2 Hz over 5 min. During tissue insertion, slight constant positive pressure (10–20 mbar) was applied to the pipette to allow constant dye perfusion of the tissue. Upon completion of the program, the pipette was retracted and the animal sacrificed after 30–60 min. In case of "double-dye" experiments, two independent electroporation experiments with fresh pipettes and solutions were performed, with approximately 15 min between experiments.

Immediately after electroporation, the initial electroporation success could already be visualized and followed directly under the two-photon microscope in vivo (Supplementary Fig. 12).

**Histological processing and data acquisition.** After transcardial perfusion with 20 ml phosphate-buffered saline (PBS) followed by 20 ml 4% paraformaldehyde, fixed brains were removed. For post-fixation, brains were kept in 4% paraformaldehyde solution for 12 h at room temperature. After extensive rinsing, tissues were embedded in a 10% gelatin block. The block was then kept in a 4% paraformaldehyde solution at 4 °C overnight. Brains were cut into 50–80 µm thick, free-floating horizontal sections on a vibratome (HM 650 V, MICROM International GmbH, Walldorf, Germany).

Slices were washed in PBS and incubated in DAPI-containing PBS solution (1:500) for 2 h at room temperature. After re-rinsing sections in PBS, they were mounted on a glass slide in VectaShield solution (Vector Laboratories Inc., Burlingame, CA, USA), coverslipped and stored at 4 °C.

Histological sections were imaged on a confocal laser microscopy system (TCS SP5, Leica Microsystems CMS, Mannheim, Germany) using a x20 glycerin immersion objective (NA 0.7). Image stacks were acquired employing an acquisition matrix of 1024 × 1024 and a z-depth of 0.6 µm. To minimize crosstalk and bleedthrough between fluorescent channels, each image frame was excited separately by each laser (405 nm for DAPI, 488 nm for fluorescein-dextran and 543 nm for TMR-dextran) and detected by only one narrow bandwidth photomultiplier in parallel. No simultaneous imaging was performed. Unless otherwise noted, all reagents were obtained from Sigma-Aldrich (St. Louis, MO, USA).

**Serial block-face scanning electron microscopy.** Fresh mouse brain slices were fixed by immersion and prepared for serial block-face scanning electron microscopy (SBEM) as described previously[66]. Two data sets were acquired on a Zeiss Merlin SEM (with Gemini II column) equipped with an automated ultramicrotome (Gatan 3View2XP) imaging with a 2 keV electron beam at a dose of 20 e$^-$/nm$^2$ and dwell times of 3 µs in high vacuum, obtaining voxel sizes of 13 * 13 * 32 nm$^3$ in x,y,z respectively across a (180 × 180) µm$^2$ field of view and throughout 40 and 200 µm in z, respectively. Data were acquired using the 3View system with DigitalMicrograph. Images were registered and normalized using Fiji[67] and later formatted for analysis in WebKnossos[68].

**Data analysis and quantification.** Acquired image data were further analyzed using Amira (Visage Imaging GmbH, Berlin, Germany), and all further quantification procedures included in the Results were performed employing custom routines in Matlab (Version R2011a, The MathWorks, Natwick, MA).

**General image alignment.** Initially, all image stacks from each experiment were manually aligned in a virtual 3D space, so that the 3D center of the *MOR174–9* glomerulus was located at the origin and the *Y*-axis cut the MCL at 90° in this central plane. The first and last image planes of the histological sections from each brain were adjusted according to continuous cellular processes in the *X* and *Y* directions and in the *Z* direction, by shifting the planes to the same level. After complete alignment of individual experiments, the entire data set was resampled and merged.

**Identification of cell soma positions.** To obtain the exact 3D positions of labeled cell somata, markers were manually placed in the center of neurons as verified by typical cell size, shape and presence of a nucleus, which was always confirmed by positive DAPI staining.

In double-dye experiments, only one dye was analyzed per cell identification session. Therefore, the operator was always blind to the second dye. This was necessary to ensure independent identification of cell somata. Only overlays with the DAPI channel were allowed.

Double-dye experiments: To quantify mutual overlap in double-dye experiments, a Euclidean distance matrix was calculated for each experiment between the identified 3D soma positions of the two dyes. If the distance between two points was <4 μm, which is less than the radius of even a small periglomerular cell, the two markers were counted as a matching double stain. This seems a good approximation, because at least two radii are expected to separate two different cells. If a marker did not show a matching partner with a Euclidean distance of <4 μm, this cell was counted as singly stained in the TMR or fluorescein channel, respectively. After acquisition of these numbers, basic statistics were calculated including mean and SD.

Total cell count analysis: Cell markers were counted and basic statistics calculated. Euclidean distances (Matlab command "pdist2") from somata to the center of the glomerulus were determined per experiment, and plotted as a histogram including SDs, to investigate the distribution of cells. Based on these data, percentage-wise radial distance borders were derived. Subsequently, each cell was attributed to its layer identity and the distance distribution was again calculated. The "layer identity" of each cell is the subregion of the MOB in which the soma is located. We defined four different subregions in our analysis: (1) GL, which is the layer comprising the olfactory glomeruli and the surrounding, tightly packed bands of cells; (2) outer EPL, which is the superficial quarter of the EPL, where cells should be separated by at least the diameter of one cell from the GL (approx. 10–15 μm); (3) inner EPL, which is the inner three quarters of the EPL, where cells should be separated by at least the diameter of one cell from the MCL (approx. 20–25 μm); and (4) MCL, which is the band of cells between the EPL and the IPL.

Transformation operation: To investigate and compare exact spatial positions between animals, a transformation operation was performed, transforming the glomerular architecture of individual experiments to one common coordinate system (Supplementary Fig. 13): First, a landmark mesh of 9 × 9 points was placed at the GL-EPL-border with its center point located in the midplane of the glomerulus on the Y-axis. The points were spaced by 30 μm in the X- and Z-directions. Additionally, the cranial pole of the glomerulus was marked. By using principal component analysis (Matlab command "princomp"), a plane was fitted to the landmark mesh and the normal passing through the origin was determined. This straight line was set as the new Y-axis by rotating the coordinate system around the X- and Z-axes. Finally, the coordinate system was rotated around the Y-axis so that the cranial pole could be preserved.

Although the exact anatomical orientation is not maintained in the transformed space because of slightly different transformation operations in each experiment, the main axes can still roughly be related to the main anatomical axes as follows: X-axis corresponds to the posterior-to-anterior direction, Y-axis to the lateral-to-medial direction and Z-axis to the dorsal-to-ventral direction.

## Cell cluster analysis of MCLPNs.

Upon completion of the transformation operation, the center-of-mass (or "common midpoint") was determined by calculating means of the three spatial coordinates. Euclidean distances from MCLPNs to the common midpoint were determined as well as distances in each of the three spatial dimensions, including mean values and SDs. The same distances were then calculated for each experiment and these individual MCL midpoints were separately and consequently compared to the common midpoint. Since MCLPNs were essentially localized on a plane parallel to the X–Z plane, the cellular density was assessed on a maximum Y projection, first counting the cell somata in bins of 20 × 20 μm along the X- and Z-axis. For smoothing, a 4 × 4 sliding frame average was performed calculating the mean of the centered bin.

After separation of MCs and dTCs (see below), the same calculations were performed for each of the two subsets separately. The only difference was that the binned cell count (bin size 50 × 50 μm) was not followed by a smoothing operation because of the low number of cells. Here, only discrete numbers were determined and plotted.

## Morphologic reconstruction of cells.

Manual and semi-automated tracing of cellular processes of MCs, dTCs, mTCs, and sTCs was achieved using the Filament Editor of the Amira software (Visage Imaging GmbH, Berlin, Germany). Detected nodes, vertices, segments and lines were exported to Matlab for further analysis. To ensure that lateral dendrites were indeed correctly identified in continuity and attributed to their respective somatic origin, we performed an independent second back-tracing from the periphery to the soma in 15 randomly chosen dendrites from 3 of our 5 experiments (i.e., 45 dendrites in total), in which complete agreement with the first tracing was achieved (Supplementary Fig. 14).

## Separation of MCs and dTCs.

First, soma size of cells was determined by fitting an elliptical shape to the interior of a cell body. The area was measured in the plane containing the largest soma size. Dendritic reconstructions of MCLPNs were imported into Matlab after truncating the apical dendrite at the most distal bifurcation node and after removal of axonal elements. Dendritic arbors consisted of a multitude of interconnected nodes or vertices giving rise to segments. Each

segment length was calculated as the Euclidean distance between the starting node and the end node of the segment. The midpoint of each segment was calculated and the relative position in the EPL determined. Each segment was then weighted by its length relative to the total dendritic length of the cell, and all the segments added so that a mean lateral dendrite position of the cell resulted. For separation of MCs and dTCs, a two-component GMM was fitted to the data of the two parameters "mean relative lateral dendrite position in the EPL" (1 = GL-EPL border, 0 = MCL) and "soma size" (Matlab command "gmdistribution.fit"). These two parameters are thought to represent principal morphological separators between the two subsets of MCLPNs[22,26,47,69,70]. The calculated posterior probabilities which correspond to "cluster membership scores" were then retrieved and plotted for each cell individually. An assumed lower boundary value of 200 μm² soma size was taken to separate "normal" MCs from "small" MCs[22].

## Morphometric analysis of MCs, dTCs, mTCs, and sTCs.

Similarly to MCs and dTCs, soma size of mTCs and sTCs was determined by fitting an ellipse to the interior of the cell body in the transverse plane containing the largest section of the soma. Mean relative lateral dendrite positions within the EPL were calculated in a fashion analogous to MCs and dTCs. To determine relative soma positions of mTCs and sTCs within the EPL, the transversal median cutting plane of a cell was identified, and the distance from the MCL to the soma and the width of the EPL were measured at the same position and the relative position calculated. Other positional parameters were determined likewise in the four cell types, including most superficial and most internal lateral dendrite extensions, most distal apical dendrite bifurcation and position of the axonal origin. Additional morphometric parameters included the number of lateral dendrites, number of secondary dendrites along the apical dendrite and distance from the axonal origin to the soma [μm].

Dendritic projections: Transformed dendritic reconstructions ("traces") from superficial TCs, middle TCs, deep TCs and MCs were imported into Matlab and relative spatial positions of the reconstructions determined: The 3D space of an experiment was first divided into small discrete bricks of 20 μm along the X- and Z-axes, which maintained full length in the Y direction (i.e., along the deep-to-superficial axis). Similar to the analysis of MCs and dTCs, midpoints of dendritic segments were then determined and each segment was assigned to one brick according to the spatial position of the midpoints. For all experiments, there were 116,230 segments assigned to 32,000 bricks. To be able first to calculate the distribution of lateral dendrites within the EPL, the local minimum and maximum dendrite extension (of any of the four cell types) within a region of 1 ± 3 bins was determined and taken as surrogate position of the MCL (minimum) and the GL-EPL-border (maximum). The difference between the two was regarded as the width of the EPL and the relative dendrite position of each segment was determined. Segments were weighted according to their length in relation to the total dendritic reconstruction length of the respective cell type, and plotted as a histogram from the MCL to the GL-EPL border. To next assess the relative position of the four trace types to each other, the mean lateral dendritic position of each of the four trace types was determined per brick, by averaging the Y positions of the segment midpoints (if there was more than one segment of the same cellular identity in one brick). Each segment was weighted by the segment length relative to the total dendrite length of the same cell type in that respective brick. Then, mean Y positions were subtracted from each other in every brick in which dendritic reconstructions from at least two cell types could be detected. The resulting differences between two traces were counted in bins of 10 μm from −250 μm to +250 μm. The counts for each of the trace types were normalized so that the integral over all counts was 1. Finally, mean values and SDs over the bin counts were calculated, and a one-sample t test was performed under the null hypothesis that the difference count was a random sample from a normal distribution with mean 0 (Matlab command "ttest"). Finally, the relative contributions of the four cell types along the (normalized) deep-to-superficial axis were calculated by first adding the absolute length of lateral dendrites from the four cell types within subregion bins of 0.05 and then dividing by the dendritic length of each individual cell type in the respective bins per experiment. The "relative average dendrite contribution" was then determined as the mean over all five experiments.

Axon collaterals: To reveal the local axonal projection pattern, first projection masks of the IPL were created (n = 3): Each individual confocal laser scanning microscopy (CLSM) image stack (40–70 μm section thickness) of the respective experiment was aligned parallel to the MCL plane, and a restricted maximum intensity Y projection (see above, corresponding to the deep-to-superficial axis) was calculated comprising only the IPL region. The resulting 2D projections of the IPL were then thresholded by the respective pixel mean of the projection and binarised. These maps were stitched together so that a binarised projection mask of the entire local IPL resulted. In these maps, only pixels from labeled axons and some vascular structures were expected to remain above 0. Image artifacts introduced by blood vessels were removed manually. Corresponding positions of MCs and dTCs, as well as the position of the projected glomerular center, were marked on this mask. To visualize preferential axonal projections, a 7 × 7 sliding frame average was performed and plotted as a heat map. To obtain a quantitative description of the correlation between axonal projections and the positions of MCLPNs, the total number of positive pixels was counted in a region of 40 × 40 μm of the original, unsmoothed mask, centered to the position of identified MCs and

dTCs of the same experiment. This count was then compared to a pixel count within a region of $40 \times 40 \, \mu m$ centered to the positions of the non-matching MCLPNs from the other two experiments as an intrinsic control. The results for the positions of all MCLPNs as well for MCs and dTCs alone were grouped, and basic statistics calculated.

**Data availability**. The data that support the findings of this study are available from the corresponding authors on reasonable request.

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

## Acknowledgements

We thank M. Kaiser and E. Stier for outstanding technical lab support. We are very grateful to A. Scherbath and G. Giese for excellent assistance with acquisition of microscopic data and to M.L. Grünbein for counting of cells and reconstruction of neuron morphologies. We are particularly grateful to the electron microscopy technical platform (Lucy Collinson and colleagues) and the scientific computing team (Wei Xing and colleagues) at the Francis Crick Institute for their continuous support in the development of the electron microscopy analysis framework. Also, we are very grateful to K. Briggman and M. Helmstaedter as well as their laboratories for help with experimental setup, staining, data acquisition, and analysis of SBEM data. Special thanks to M.R. Angle who has always been a constant source of fruitful discussions and visionary ideas. We also thank T. Kuner for providing us with Thy1-CLM-transgenic animals and W. Denk for support throughout. This work was supported by the Francis Crick Institute, which receives its core funding from Cancer Research UK (FC001153), the UK Medical Research Council (FC001153), and the Wellcome Trust (FC001153); the Max-Planck-Society and by the UK Medical Research Council (grant references MC_UP_1202/5). A. T.S. is a Wellcome Trust Investigator (110174/Z/15/Z).

## Author contributions

D.S., M.K., and A.T.S. conceived and designed the study. D.S. and A.T.S. wrote the manuscript. D.S. and M.K. performed the electroporation experiments. D.S. analyzed the results. C.F. implemented and performed FEM modeling and FEM data analysis. C.B. designed and performed the EM experiments with input from D.S., M.K., I.W., T.W.M., and A.T.S. I.W. and C.S. analyzed the EM data. T.C. provided transgenic MOR174–9 animals. M.K., C.F., C.B., and T.C. critically revised the manuscript and aided with the design and analysis of experiments. All authors discussed the results and commented on the manuscript.

## Additional information

**Competing interests:** The authors declare no competing financial interests.

