## [Peer Review File · Nature Communications]

Reviewers' comments:

Reviewer #1 (Remarks to the Author):

Review of Schaefer

This manuscript describes a new approach to localized electroporation that allows for dense reconstruction of local neural circuits. The approach, which depends on a novel perforated tip electrode represents a significant advance that may facilitate labeling approaches in a variety of brain areas. The authors apply this technology to obtain near-complete labeling of all neurons affiliated with an identified olfactory bulb glomerulus. They demonstrate a remarkable degree of segregation in dendritic arborizations in different types of projection neurons. They also demonstrate a high degree of targeting of axonal projections. The value of the manuscript stems from the fact that it represents both a technical accomplishment and a set of novel biological findings about the circuitry of the olfactory bulb. I have comments that would strengthen the paper that focus on both of these aspects of the work.

An estimate of 80% completeness is likely sufficient for many analyses, including those performed here, but it would be useful to know whether there were any apparent biases in the cells that were not doubly labeled in experiments described in figure 3. Were the non-doubly labeled cells likely to be located close to the glomerulus or far? Were they more likely to be small cells or large cells?

Some additional information on how the robustness of tracings was evaluated should be provided. In the densely labeled tissue near the glomerulus especially, how were nearly touching dendrites evaluated to ensure that continuous segments were not mis-identified. This is especially true across physical sections. I understand that complete reconstructions were not performed in the traditional sense, but interpretation of data from e.g. figure 6 are dependent on knowing the somatic origin of a particular dendritic segment.

Authors should comment more directly on whether cells having axons, but not dendrites in the targeted glomerulus were labeled by their electroporation. Did they label cell types - such as deep or superficial short axon cells - that appeared only to have axons in the glomerulus labeled?

More detailed analysis of the degree of overlap in lateral dendrites of the identified cell types would be of interest. From the images the overlap looks remarkably minimal. However, some additional assessment such as the relative density of dendritic length in the EPL as a function of distance from the mitral cell layer would be helpful in giving a better appreciation of the results. In figure 6 are the histograms from one experiment? Or assembled across experiments? In my opinion, what would be most useful for integration with future studies would be if - for a given fractional distance between MCL and GLL the authors could plot the relative fraction of dendrites originating from a given cell type. For example - at 50 μm from the MCL one might see that 85% of dendritic length comes from MCs 10% from dTCs 5% from mTCs and 0% from sTCs. Ideally this would be plotted as a function of the radial distance from the glomerulus because dendrites may become more "mixed" at sites more distant from the glomerulus. This cannot be easily extracted from the data as presented, but would be very helpful for understanding and interpreting the data presented.

Authors should comment on the viability of the electroporated neurons. In vitro experiments would allow direct recordings to be used, but other less direct measures would also be helpful. They should also provide data on whether the labeling was sufficient to allow identification of some of these cells in vivo under the 2P microscope.

Minor:

Authors should cite recent paper of Liu and Urban that uses standard electroporation of the M72 glomerulus and reaches some similar conclusions to the present MS related to position and number of mitral cells.

Lines 110+ The phrase "compromised volume" is unclear. Is this the volume of the damaged tissue?

Line 173+ The statement "In our definition, every directly glomerulus-affiliated, lateral dendrite-bearing neuron was considered a TC." Is somewhat unclear. I suggest "We define as tufted cell as every neuron that is not a mitral cell that has a primary dendrite targets a glomerulus and also has lateral dendrites."

Reviewer #2 (Remarks to the Author):

Schwartz et al., in their manuscript "Architecture of a genetically targeted mammalian glomerular domain revealed by a novel volume electroporation technique using nanoengineered microelectrodes" describe an elegant novel method for spatially targeted labeling of neuronal circuits and put it to work to characterize the relative distribution, spatial location and local projection features of neurons innervated by one genetically defined glomerulus (MOR174) – homotypic or sister cells. The technique presents practical advantages upon classical electroporation methods, allowing larger spread, yet focal targeting of the labeling agent, while minimizing the destruction of neuronal tissue. In addition, the authors identify layered structure of lateral dendrites of different classes of TCs and MCs, as well as potentially interesting clustering of TC axonal projections and MC cell bodies from the same glomerulus.

However, I have several major concerns regarding the applications of this technique, in the current form of the manuscript, for identifying neurons associated with an individual glomerulus that should be addressed before publication.

1) In my opinion, the data presented does not clearly demonstrate that ~80% of neurons associated with the target glomerulus are indeed labeled. The authors show that two successive electroporation events with two different color dyes result in 80% overlap in labeled cells (Figure 3), and furthermore take advantage of Thy1-CLM line that sparsely labels MC & TCs to support their claim. Yet, 80% overlap in labeling via successive electroporation events may not equate 80% labeling of all neurons associated with the glomerulus; instead it only shows the reliability of the method (on average ~80% same neurons are marked). Furthermore, it is unclear what fraction of MC, TC & ET cells are genetically targeted to express CLM in the Thy1-CLM mouse line, and hence remains uncertain that the technique labels most of the neurons receiving input from this glomerulus. In fact, the authors comment in the Discussion that specific juxtglomerular cells well documented by previous publications are not labeled using this approach, which raises some doubt about the efficiency of the technique.

2) The authors make a strong point regarding the reproducibility of MOR174 glomerulus sister MC and TC cluster spread across animals (<100 μm), which is proposed to be smaller even than the positional variability of glomeruli (but see Soucy et al., Nat. Neurosci, 2009). The implications of this finding are not immediately obvious, given that the range of integration of MCs and TCs is dependent not necessarily on the position of their cell bodies, but on the length, orientation and connectivity patterns of their lateral dendrites. Furthermore, the authors comment that the spread of sister MC bodies is wider compared to TCs along D-V axis and, but it remains unclear whether this spread has clear

functional consequences. More generally, it would be interesting to determine whether the same set of results, as well as for example the frequency of different classes MC, TC, JGC applies to different glomeruli.

3) The clustering of TC axons onto mitral cell bodies and is proposed as strongly indicative of TC-GC-MC connections, but no direct evidence is given to support this proposed interpretation. Given that this result is presented as a major finding of the manuscript, as the anatomical basis of a novel parallel circuit, additional evidence is needed to support the claim.

Reviewer #3 (Remarks to the Author):

In the manuscript by D. Schwarz et al., the authors describe a novel method using nano-engineered electroporation microelectrodes to label neuronal networks associated with single olfactory glomeruli. They show that this new method allows for improved current density, improved current distribution, and electroporation efficacy by allowing a greater volume of material to be electroporated through multiple distributed release sites, with less damage than previously described local electroporation approaches. The authors applied this technique to characterize the neurons associated with a single glomerulus, quantifying the associated cell-types by allocating them into different layers of the olfactory bulb, and further characterizing the projection patterns of the labeled cells. Additionally, from these data the authors further propose a novel feedforward circuit from tufted cells that ultimately affects mitral cells. Overall, this paper presents a novel technique for the study of isolated neuronal networks that could be broadly applied for the detailed study of brain circuits. The technique seems to be very robust, and the manuscript is a great fit for the journal by addressing a few points.

Comments:

1. It would be useful to include data, or at least describe differences in electroporation volume, and/or labeling efficiency using different patterns or numbers of release sites. This would nicely augment the modelling data.
2. Although it is appreciated that a focus of the analysis was to categorize projection neurons to glomeruli, the olfactory bulb layers contain heterogeneous cell populations. It would be informative to include some marker analysis to definitively identify TCs vs. MCs vs. SMCs vs. interneurons. The criteria are not always clear in the text of how the authors parse out the different projection neuron subtypes aside from anatomy.
3. Given the claim "this association strongly suggests a feedforward TC to GC to MC pathway", it now seems almost imperative for provide more than anatomical analysis to provide cell type specificity. The manuscript would benefit from either building upon this speculation with data, or deemphasizing (or removing) this point.
4. Practical discussion of other applications using this platform would be welcomed. For example, does this approach work to electroporate local neurons with DNA constructs?
5. The supplementary figures are listed out of order throughout the text. These should be chronologically labeled as the figure data is described.

Reviewer #1 (Remarks to the Author):

Review of Schaefer

This manuscript describes a new approach to localized electroporation that allows for dense reconstruction of local neural circuits. The approach, which depends on a novel perforated tip electrode represents a significant advance that may facilitate labeling approaches in a variety of brain areas. The authors apply this technology to obtain near-complete labeling of all neurons affiliated with an identified olfactory bulb glomerulus. They demonstrate a remarkable degree of segregation in dendritic arborizations in different types of projection neurons. They also demonstrate a high degree of targeting of axonal projections. The value of the manuscript stems from the fact that it represents both a technical accomplishment and a set of novel biological findings about the circuitry of the olfactory bulb. I have comments that would strengthen the paper that focus on both of these aspects of the work.

We thank the reviewer for their concise summary of our work and their very encouraging comments. We have implemented all suggestions through new experiments and new data analysis as detailed below.

An estimate of 80% completeness is likely sufficient for many analyses, including those performed here, but it would be useful to know whether there were any apparent biases in the cells that were not doubly labeled in experiments described in figure 3. Were the non-doubly labeled cells likely to be located close to the glomerulus or far? Were they more likely to be small cells or large cells?

We thank the reviewer for the comment and analysis suggestions. We now include a subanalysis of the four subregions I) glomerular layer II) outer EPL III) inner EPL and IV) MCL in our figure 3 as new subfigure g) (Reproduced here as Figure R1). Indeed, there is some difference between the layers but it does not seem to be

Figure R1 Double in vivo electroporation (part of Fig. 3 in the revised manuscript)

(f) Individual data plot of three double-labeling experiments (Expts. A-C). For each experiment, the left bar shows the number of TMR-positive cells and the right bar the number of fluorescein-positive cells. Each bar is subdivided into doubly-labeled fraction of cells (brown) and singly-labeled fraction (red for TMR and green for fluorescein). In **(g)**, the fraction of doubly-labeled, fluorescein-positive cells is shown for different layers. Blue bar and circle indicate mean values and standard deviation.

distance dependent as MCL cells which are located farther away from the electroporation site seem to be doubly labeled highly reliably. In fact, the large MCL cells are double-labeled almost completely ($96.7 \pm 5.8\%$), whereas the smaller cells (which are located closer to the electroporation site) are lost to a higher degree on the second electroporation epoch (recovery rate of $78.8 \pm 3.8\%$). As the GL cells far outnumber the MCL cells, the overall number we reported previously ($81.6 \pm 1.2\%$) is almost identical to the GL value. This is now presented as the new Figure 3g and discussed on p. 5 and p. 10 of the revised manuscript.

Some additional information on how the robustness of tracings was evaluated should be provided. In the densely labeled tissue near the glomerulus especially, how were nearly touching dendrites evaluated to ensure

that continuous segments were not mis-identified. This is especially true across physical sections. I understand that complete reconstructions were not performed in the traditional sense, but interpretation of data from e.g. figure 6 are dependent on knowing the somatic origin of a particular dendritic segment.

We completely agree that correct continuous identification of dendrites and their respective somatic origins is essential to the reliability of our quantitative analyses. Except for very rare cases, we did not find ourselves confronted with major problems when tracing dendritic processes – even across physical sections – most probably due to sufficient sparsity of the labelling within the EPL. To better evaluate the robustness of our tracings, we now performed an additional independent back-tracing from 15 randomly chosen dendrites in 3 of our 5 experiments, i.e. 45 dendrites in total. These 15 dendrites per experiment (5 within the upper EPL, 5 within the central portion of the EPL and 5 of the inner part of the EPL) were traced backwards from the very periphery of our sections to the somatic origin and were then compared to the initial tracing of our analysis. In all cases,

Figure R2: Robustness of dendritic tracing (Supplementary Fig. 14 of the revised manuscript)

Example back-tracing experiment with the retrace shown by magenta dots. White lines represent dendritic trajectories of the original dataset, red dots show soma locations within the MCL while blue dots indicate soma locations in the superficial EPL. (a) shows a lateral (y-z) view with a detail overlay of the original data stacks outlining physical break-points. (b) and (c) indicate the corresponding orthogonal planes x-y and x-z, respectively.

the same somatic origin was identified and the dendritic trajectory matched the initially identified course of the dendrite. The dendrites of the upper EPL portion had 3.87 ± 1.86 , the dendrites of the middle EPL part had 2.87 ± 0.88 and the dendrites of the inner EPL had 3.93 ± 1.12 physical breakpoints. Dendritic length was $598.3 \pm 185.8 \mu\text{m}$, $485.0 \pm 114.0 \mu\text{m}$ and $640.3 \pm 231.9 \mu\text{m}$, respectively. Interestingly, consistent with our EPL analyses, lateral dendrites of the superficial portion belonged to sTC in 14/15 cases and to an mTC in one case. Dendrites from the middle portion originated from mTC in 3/15 cases and the rest belonged to MCL neurons – as did all dendrites from the inner part of the EPL. An example experiment is shown in figure R2 in which the retrace is shown by magenta dots. White lines represent dendritic trajectories of the original dataset, red dots show soma locations within the MCL while blue dots indicate soma locations in the superficial EPL.

Thus, we conclude that following labeled dendrites and axons across physical sections is possible reliably for sufficiently sparse labeling (like in the cases analysed here in the EPL, and e.g. Figures 5-7 of the main paper). Dis-entangling dendrites of e.g. different periglomerular cells within the glomerulus might be more challenging and could benefit from other (e.g. superresolution) imaging techniques and thus possibly other dyes.

The above quantifications and the conclusions we draw from them are now discussed on p. 22, of the revised manuscript and illustrated in Supplementary Figure 14.

Authors should comment more directly on whether cells having axons, but not dendrites in the targeted glomerulus were labeled by their electroporation. Did they label cell types - such as deep or superficial short axon cells - that appeared only to have axons in the glomerulus labeled?

Thank you very much for raising this point. In fact, we do find some neurons around the glomerular layer which can be located some glomeruli away from the electroporation site and these neurons extend slender, axon-like processes into the targeted glomerulus like the one shown in the Figure R3a which may correspond to an “oligoglomerular Tyrosine-hydroxylase positive neuron” (4, 5). Besides, olfactory nerve fascicles which basically contain olfactory sensory nerve axons can indeed be electroporated successfully und a glomerulus be labeled

Figure R3 Electroporation of axons and axon-like processes (a) Successfully electroporated neuron close to a neighboring glomerulus featuring axon-like processes (red arrow), presumably corresponding to an “oligoglomerular Tyrosine-hydroxylase positive neuron”. **(b)** Olfactory nerve electroporation with subsequent labeling of the corresponding glomerulus. Scale bar is 50 μm .

(Figure R3b). However, while the analysis of TC/MC lateral dendrites seems straight forward due to easy identification of these structures, robust assessment of the identity of neurites for GL neurons seems more challenging: Since morphological features of glomerular layer neurons can be diverse and dendritic and/or axonal processes may resemble each other – especially in or very close to the intensely labeled glomerulus itself – we do not feel confident enough to provide a reliable quantitative judgement about the exact identity, frequency and morphological features of these cells.

We now discuss the differential labelling of axons and dendrites on p. 11 of the revised manuscript.

More detailed analysis of the degree of overlap in lateral dendrites of the identified cell types would be of interest. From the images the overlap looks remarkably minimal. However, some additional assessment such as the relative density of dendritic length in the EPL as a function of distance from the mitral cell layer would be helpful in giving a better appreciation of the results. In figure 6 are the histograms from one experiment? Or assembled across experiments? In my opinion, what would be most useful for integration with future studies would be if - for a given fractional distance between MCL and GLL the authors could plot the relative fraction of dendrites originating from a given cell type. For example - at 50 μm from the MCL one might see that 85% of dendritic length comes from MCs 10% from dTCs 5% from mTCs and 0% from sTCs. Ideally this would be plotted as a function of the radial distance from the glomerulus because dendrites may become more “mixed” at sites more distant from the glomerulus. This cannot be easily extracted from the data as presented, but would be very helpful for understanding and interpreting the data presented.

We thank the reviewer for this excellent suggestion. In the original manuscript, we provided the group data (across $n=5$ exps) dendritic distribution of each of the four cell types individually in figure 6 c-f. The relationship to each other cell type was shown in supplementary figure 6.

We now include an additional subfigure 6g) in which we show the average dendritic contribution (across $n=5$ exps) as a function of EPL depth (MC=red, dTC=cyan, mTC= yellow, sTC=magenta), reproduced here as Figure R4 and added a corresponding comment in our results section (p. 8).

Figure R4 Segregated dendritic bands within the EPL (part of Fig. 6 of the revised manuscript)

Histogram plots showing the average distribution of the relative position for lateral dendrites within the EPL from the four cell types, in bins of 0.05 from the MCL (0) to the GL-EPL border (1) ((c) = sTC, (d) = mTC, (e) = dTC and (f) = MC). Black vertical bars indicate \pm standard deviation. Overall mean dendritic position for each cell type is shown \pm standard deviation (number and coloured point and horizontal bar above each histogram). (g) Average relative dendrite contribution from the four cell types (magenta = sTC, yellow = mTC, cyan = dTC, red = MC) per subregion of the normalized EPL width in bins of 0.05. Right y-axis showing the contribution as absolute distance with a typical EPL width of 180 μm .

However, the EPL-axis was normalized from 0 to 1 in this analysis as EPL thickness could vary considerably between animals (between 150 and 210 μm). and sometimes even within one experiment due to a varying curvature (of the layers) of the bulb. Therefore, we do not think that a similar plot as function of radial distance from the glomerulus across several experiments/animals would yield sufficiently conclusive results to be presented. An impression of the absolute distances, however, can be obtained when setting the normalized thickness of the EPL to 180 μm which represents the most typically encountered width in our study cohort. Another indication that intermingling does not take place even at distant sites to the glomerulus is the observation that the distant position of random lateral dendrites strictly respects the 1/3 vs 2/3 partitioning of the EPL between sTCs and MCL neurons (Fig. R2).

Authors should comment on the viability of the electroporated neurons. In vitro experiments would allow direct recordings to be used, but other less direct measures would also be helpful. They should also provide data on whether the labeling was sufficient to allow identification of some of these cells in vivo under the 2P microscope.

Figure R5 *Two-photon visualization of electroporated neurons in vivo (Supplementary Fig. 12 of the revised manuscript)*

Two-photon projection of the MOR174-9 glomerulus labeled by local electroporation of Alexa594 hydrazide. Scale: 50 μ m

We could indeed visualize electroporated neurons readily *in vivo* (Figure R5) and now include this in the revised manuscript as Supplementary Figure 12 and in our Method section (p. 18). We furthermore discuss viability in the revised manuscript on p. 10. Generally, viability is thought to primarily depend on the current densities employed. As we have reduced current densities due to the distribution of current across the set of nanofabricated holes, we would conclude that viability should be at least comparable to previous approaches. Nevertheless, the reduced electroporation efficacy on the second attempt compared to the first suggests that at least transiently membrane integrity of electroporated neurons might be compromised in particular for small neurons (as suggested before (6, 7) cited in the revised manuscript on p. 10).

Minor:

Authors should cite recent paper of Liu and Urban that uses standard electroporation of the M72 glomerulus and reaches some similar conclusions to the present MS related to position and number of mitral cells.

Thank you for pointing out this omission – we now discuss and compare the work of Liu and Urban to ours on p. 11 of the revised manuscript.

Lines 110+ The phrase “compromised volume” is unclear. Is this the volume of the damaged tissue? We have rephrased this section on p 5 in the revised manuscript. We indeed wanted to refer to the tissue volume at risk of possible irreversible electroporative damage according to (8).

Line 173+ The statement “In our definition, every directly glomerulus-affiliated, lateral dendrite-bearing neuron was considered a TC.” Is somewhat unclear. I suggest “We define as tufted cell as every neuron that is not a mitral cell that has a primary dendrite targets a glomerulus and also has lateral dendrites.” Thank you for this suggestion which we have implemented in the revised manuscript on p. 7.

Reviewer #2 (Remarks to the Author):

Schwartz et al., in their manuscript "Architecture of a genetically targeted mammalian glomerular domain revealed by a novel volume electroporation technique using nanoengineered microelectrodes" describe an elegant novel method for spatially targeted labeling of neuronal circuits and put it to work to characterize the relative distribution, spatial location and local projection features of neurons innervated by one genetically defined glomerulus (MOR174) – homotypic or sister cells. The technique presents practical advantages upon classical electroporation methods, allowing larger spread, yet focal targeting of the labeling agent, while minimizing the destruction of neuronal tissue. In addition, the authors identify layered structure of lateral dendrites of different classes of TCs and MCs, as well as potentially interesting clustering of TC axonal projections and MC cell bodies from the same glomerulus.

However, I have several major concerns regarding the applications of this technique, in the current form of the manuscript, for identifying neurons associated with an individual glomerulus that should be addressed before publication.

We want to thank this reviewer very much for their critical assessment of our manuscript. We have addressed the points made with substantial new data, new analysis and added discussion as described in detail below and are convinced that the manuscript has improved considerably thanks to that.

1) In my opinion, the data presented does not clearly demonstrate that ~80% of neurons associated with the target glomerulus are indeed labeled. The authors show that two successive electroporation events with two different color dyes result in 80% overlap in labeled cells (Figure 3), and furthermore take advantage of Thy1-CLM line that sparsely labels MC & TCs to support their claim. Yet, 80% overlap in labeling via successive electroporation events may not equate 80% labeling of all neurons associated with the glomerulus; instead it only shows the reliability of the method (on average ~80% same neurons are marked). Furthermore, it is unclear what fraction of MC, TC & ET cells are genetically targeted to express CLM in the Thy1-CLM mouse line, and hence remains uncertain that the technique labels most of the neurons receiving input from this glomerulus. In fact, the authors comment in the Discussion that specific juxtglomerular cells well documented by previous publications are not labeled using this approach, which raises some doubt about the efficiency of the technique.

The reviewer raises an important point. As electroporation is a biophysical technique that should predominantly depend on field distributions relative to neuronal membranes (and possibly the size and orientation) rather than surface molecule distribution or other "biological" variables, that argues for less bias than other techniques such as viral infections. It is, however, correct that the "independent" retest-approach primarily probes the reliability of our method. Thus, to further back up our point, we have performed several sets of additional experiments and analysis:

- (i) As described also in responses to reviewer 1, reproducibility between repeated electroporation epochs indeed varies for different cell types: We have now analyzed electroporation reliability for cells in different layers. There is some difference between the layers but it does not seem to be e.g. distance dependent as MCL cells which are located farther away from the electroporation site seem to be doubly labeled highly reliably. In fact, the large MCL cells are doubly labeled almost completely ($96.7 \pm 5.8\%$), whereas the smaller cells (which are located closer to the electroporation site) are lost to a higher degree on the second electroporation epoch (recovery rate of $78.8 \pm 3.8\%$). As the GL cells far outnumber the MCL cells, the overall number we reported previously ($81.6 \pm 1.2\%$) is almost identical to the GL value. As our further analyses primarily deal with the MC/TC domain, we conclude from this subanalysis that we indeed cover almost every single of these neurons. This is now presented as the new Figure 3g (reproduced above as Figure R1) and we now emphasize this point in the revised manuscript on p. 5 and p 10.

- (ii) In order to get an independent view at cell populations present in and around the glomerulus in a parallel study we have recorded and filled large numbers (>90) of JGCs in a “unbiased” way (at least no explicit preselection based on soma location, size or genetic markers) during *in vitro* whole-cell patch recordings. Cluster analysis of these reveal several distinct morphological groups, partially mirroring previously described cell types. Re-evaluating our existing and complementing new electroporation data, we could indeed find examples from each of these cell groups (Figure R6). While this is certainly no proof for the technique not missing any cell type it strongly suggests that there is no obvious bias apart from a presumed reduced efficiency for smaller neurons in general. This, however, may be linked to the volumetric range of our proposed electroporation settings which lies below the whole volume of the considered target circuit (40–50 μm radius). As explained in our revised discussion (p. 11), this was necessary in order “not to compromise the required specificity when handling the pipette at the microscale. Especially small cellular processes passing only very peripherally through the glomerulus might therefore still be missed by electroporation”

Figure R6 Classes of glomerular layer neurons

Column A and B: Figure from Tavakoli, Schaefer, Kollo in preparation: Morphology-based dendrogram showing 7 classes of GL neurons (A-G). Right column: Corresponding examples of electroporated glomerular layer neurons (red arrows, classes A-G).

- (iii) Notably, while we initially believed to have missed so-called “juxtglomerular association neurons” (4, 5), a cell type which is represented in our overview of JGCs as class F, we could indeed identify some of such neurons in our electroporation experiments after thorough re-evaluation if their soma could be located in neighbouring glomeruli (i.e. < 300µm distant). More often, however, we could find relatively straight, unbranched axon-like processes traversing several glomeruli resembling extensions of such juxtglomerular association neurons without being able to locate their somatic origin (Fig. R7). This again points to the fact that certain cell types of the glomerular network are not *per se* “unelectroporatable” but that retrograde filling of small processes may not be as efficient in our experimental regime. We clarify this point now in our discussion p. 11 and include Fig. R7 as a new Supplementary Figure 9.

Figure R7 Successfully electroporated thin processes of putative “juxtglomerular association neurons” in distant glomeruli (Supplementary Fig. 9 from the revised manuscript)

Red arrows indicate fine transglomerular axon-like processes resembling those of putative juxtglomerular association neurons (1, 2) in glomeruli distant to the original electroporation site. Scale bars 100 µm.

- (iv) While neuron number likely varies significantly between glomeruli, many studies have tried to provide quantitative morphometric descriptions of different elements in the olfactory bulb across species (Table R1, (9-20)). However, most of these studies used global quantification methods in which the total population of glomeruli and/or the overall number of certain cell types were estimated. The composition of the glomerular domain was then calculated as an average ratio of cells per glomerulus. Moreover, cell type identification has only been based on bulbar layer identity of the cells and not on morphological parameters as in our study (15). More specific quantitative approaches (Table R1, (20-22)) have only become feasible in recent years with the arrival of targeted electroporation.
- Since no systematic assessment of the quantitative extent of this technique has been undertaken to date, the exhaustiveness of the method has been unclear but numbers between 7 and 16 MCL cells per glomerulus were reported in these studies. Other cell types have not yet been assessed quantitatively by this or a similarly direct technical approach.
- Importantly, the total numbers of cells as well as the numbers of MCL cells per glomerulus of the earlier, global estimates and the specific targeted approaches differ substantially, i.e. at least by a factor of two. This difference might be attributable to an incomplete delineation of the population of glomerulus-associated neurons by the targeted electroporation approach, but relevant interobserver variability in the assessment of global estimates may also play a role. However, an interesting recent study (14) provided convincing evidence that earlier global estimates in mice must be challenged due to an about twofold higher re-estimation of the total number of glomeruli

per bulb compared to earlier studies (9, 10, 12) when using a new, potentially more rigorous approach based on immunohistochemistry that allows for reliable detection of small glomeruli. Such glomeruli had most likely been neglected in earlier studies resulting in a systematic underestimation of the number of glomeruli (9, 10, 12). Thus, existing global estimates of the number of cells per glomerulus are likely to be inaccurate and a ‘gold standard’ to achieve a reliable quantitative description of the neuronal elements of the glomerular circuitry does not exist. Additionally, some cell types of the bulb such as short axon cells and other local interneurons do not extend any cellular process into the glomeruli but would likely be contained in average neuron counts.

Taking these limitations into account, we feel confident that the average cell number per MOR174-9 glomerulus of around 200 cells we find in our electroporation data (taking into account a ‘missed’ number of 20 %) might be a most direct estimate of cell number. This is lower than estimated previously (15) but their reported figure of 441 cells was based on an earlier total number of glomeruli (10) and when adjusted by the most recent count by Richard *et al.* (14), their global estimation lies in a similar range (~ 220 cells). We now address this point more explicitly in our discussion on pp. 10/11.

The impact of animal age may also have to be taken into account here: PGs are continuously replaced by adult born neurons in the SVZ (23). While earlier studies have claimed neuronal stability with a balanced turnover rate of neurons in the GL (24), it was reported more recently that at least the subpopulation of dopaminergic PGs showed a relatively strong increase with age (25). The authors of that study did not explicitly address the question whether this finding was due to a dynamic remodelling of subpopulations within an overall stable total population, or whether this was hinting at an overall growing population. Also, environmental sensory enrichment did enhance proliferation of neuronal precursors in the SVZ already (26) as well as survival of newborn neurons (27). Therefore, the number of around 200 glomerulus-associated cells in the relatively young animals used here (< 2 months) might increase further with age and olfactory experience.

Overview of quantitative morphometric studies in the olfactory bulb

	Approach used	Species	Total number of glomeruli	GL	EPL/TCs ¹	MCL/MCs ¹
Allison & Warwick (1949)	Global	Rabbit	1901± 28	N/A	59 ± 1	23 ± 2
Meisami & Safari (1981)	Global	Rat	2737 ± 28	N/A	32 ± 1	26 ± 1
Benson et al. (1984)	Global	Mouse	N/A	N/A	N/A	38355 ± 1428 ²
Panhuber et al. (1985)	Global	Rat	N/A	N/A	N/A	70880 ± 6210 ² 48240 ± 2010
Royet et al. (1998)	Global	Rabbit/Rat	6297 ± 343 (rabbit) 4187 ± 368 (rat)	N/A	N/A	9.5 ± 1 (rabbit) 13 ± 2 (rat)
Nawroth et al. (2007)	Global	Mouse	1810 ± 41 (Royet et al. , 1988)	100	60	25
Parrish-Aungst et al. (2007)	Global	Mouse	1810 ± 41 (Royet et al. , 1988)	441 ± 28	33 ± 5	25
Richard et al. (2010)	Global	Mouse	3859 ± 300-350	N/A	N/A	9 ± 1
Sosulski et al. (2011)	Specific	Mouse	N/A	N/A	N/A	9 ± 1
Ke et al. (2013)	Specific	Mouse	N/A	N/A	N/A	15.8 ± 4.3
Liu et al. (2016)	Specific	Mouse	N/A	N/A	2.9 ± 0.9 (ctrl) 6.6 ± 1.4 (mint)	6.8 ± 1.1 (ctrl) 10.0 ± 1.1 (mint)

¹In the studies, the terms TCs and MCs were used as synonyms for cells of the EPL and MCL, respectively

²In these studies, only total numbers of MCs per bulb were determined

Table R1 Overview of quantitative morphometric studies in the olfactory bulb

Taken together, while we acknowledge that a ‘ground truth’ neuronal number per glomerulus does not exist to date and our independent retest-approach provides only a simplifying assumption of completeness, the numbers we provide are highly consistent with all indirect measures and theoretical considerations which we can find in the literature.

We do discuss this important point in the discussion on pp. 10/11 of the revised manuscript as well as in substantially more detail in a new supplementary discussion which we reference on p. 11.

2) The authors make a strong point regarding the reproducibility of MOR174 glomerulus sister MC and TC cluster spread across animals (<100 μm), which is proposed to be smaller even than the positional variability of glomeruli (but see Soucy et al., Nat. Neurosci, 2009). The implications of this finding are not immediately obvious, given that the range of integration of MCs and TCs is dependent not necessarily on the position of their cell bodies, but on the length, orientation and connectivity patterns of their lateral dendrites. Furthermore, the authors comment that the spread of sister MC bodies is wider compared to TCs along D-V axis and, but it remains unclear whether this spread has clear functional consequences. More generally, it would be interesting to determine whether the same set of results, as well as for example the frequency of different classes MC, TC, JGC applies to different glomeruli.

a

Firstly, we apologize for the misunderstanding. We did not want to suggest that the position of the glomerulus itself was reproducible but that the position of MCs and TCs *relative to the glomerulus* is reproducible. In order to assess the reproducibility of glomerular position we dissected whole OBs from 5 OR174-GFP animals and determined the location of the fluorescent glomerulus in the A-P, M-L axis. Indeed, position was highly variable, especially when not normalized to OB size (Location of dorsal MOR174 in 6.1 week-old mice (3 males, 2 females, n = 10 bulbs pooled together): A-P: between 77.88 and 89.81% anterior. Mean 83.49%, SEM 1.126%. CI 95% = [80.94, 86.04]; in μm : between 317.5 and 980.5 μm anterior. Mean 663.6 μm , SEM 63.79 μm . CI 95% = [519.3, 807.9]; M-L: between 51.35 and 61.7% lateral. Mean 56.2%, SEM 0.969%. CI 95% = [54.01, 58.39]; in μm : between 1090 and 1346 μm lateral. Mean 1219 μm , SEM 21.74 μm . CI

b

c

Figure R8 Positional variability of the MOR174-9 glomerulus (a) shows top-view fluorescent images of five dissected olfactory bulbs. (b) relative position of the MOR174-9 along a-p and m-l axes. (c) absolute dimension of the olfactory bulb size in μm .

95% = [1170, 1269]; Dimensions of these OBs: Width: between 2059 and 2267 μm . Mean 2170 μm , SEM 19.52 μm . CI 95% = [2126, 2215] Length: between 2370 and 2918 μm . Mean 2736 μm , SEM 49.36 μm . CI 95% = [2624, 2847]). While the methodology is relatively crude compared to the detailed 3-dimensional reconstructions of glomerular locations performed recently by the Mombaerts group (28), it is indeed very consistent with both this recent publication and the functional imaging work mentioned by the reviewer. We now include this data on p. 16 and emphasize in the revised manuscript that the stereotypy we observe with respect to MC and TC location is *relative to the glomerulus* with glomerular position being variable between animals on p. 11.

Secondly, we had indeed refrained from speculating about the functional consequences of the anatomical observation we have made in the original manuscript. While the extent of lateral dendrites indeed allows a MC to excite GCs at distant positions and the specific direction and length of the lateral dendrite will be possibly a much more decisive factor than the MC soma position, this is not so much the case for receiving inhibitory input. Perisomatic inhibition is by far more effective than distal inhibition. In fact, inhibition onto lateral dendrites might predominantly affect AP propagation and possibly MC-GC excitation and have a much more limited effect on regulating MC discharge. Here, perisomatic synapses are particularly powerful. Thus, MC soma position will determine which population of GCs can control MC firing (29) which might represent an important link to the earlier proposed “olfactory columns” (30-32). Moreover, the tight clustering of MCL neurons is well in line with earlier observations by Luo and Katz (33) as well as anatomical observations that sets of sister MCs were separated by less than 120 μm (34). Of course, it is less clear how GC position in itself determines GC excitation considering the seemingly non-topographic nature of centrifugal inputs. Nevertheless, the position of MC somata can have profound effects on how they are inhibited. Interestingly, our observed preferential displacement of MCs follows the direction along which major transitions from different OR-zones occur (35-37) which may allow for efficient sampling and integration from a spectrum of various kinds of olfactory information

While we still think that this is largely speculation we feel encouraged by the reviewer’s comments to expand our discussion regarding potential implications on p.12 of the revised manuscript.

Figure R9 MCL neuron analysis in a dorsal non-GFP glomerulus compared to the MOR174-9-GFP glomerulus
(a) Spatial positions of MCL neurons (dots) along D-V and P-A axis (similar to Fig. 5) of the non-GFP glomerulus (magenta) in comparison to the positions of the MOR174-9-GFP (red, green, blue, cyan and black indicate experiment identity). Triangles indicate MCL neuron midpoints and blue circle at the origin indicates respective glomerular centre. **(b)** Distance of the MCL midpoint to the glomerular centre. Yellow cross and bar indicate mean \pm standard deviation, $** = p < 0.01$. **(c)** Distance distribution of MCL neurons of MOR174-9-GFP glomerulus and non-GFP glomerulus to each corresponding MCL midpoint. Yellow and magenta cross and bar indicate mean \pm standard deviation, $n.s. = p > 0.05$. **(d)** Distance distribution of MCs and dTCs, respectively, of MOR174-9-GFP glomerulus and non-GFP glomerulus to each corresponding MC and dTC midpoint. Red, cyan and magenta cross and bar indicate mean \pm standard deviation, $n.s. = p > 0.05$

In order to assess the distribution of MCL neurons in another glomerulus we have now performed electroporation experiment in a dorsal, non-MOR174 glomerulus for comparison (Figure R9). In this glomerulus, the Euclidean distance of the MCL neuron midpoint to the origin in the centre of the respective electroporated glomerulus is significantly greater (335 μm) than the common MCL midpoint of the MOR174-9 glomerulus (277 \pm 11 μm , two-sided t-test $p = 0.008$, C.I. -91.7; -25.5 μm) so this distance may depend on the actual position of a

glomerulus. Within the glomerular network, however, we find a similar distance distribution of the MCL neurons around their common midpoint ($129.2 \pm 78.2 \mu\text{m}$) as in case of the MOR174-9 Glomerulus ($101.1 \pm 65.8 \mu\text{m}$, n.s., $p=0.34$, C.I. $-86.8;30.5$). This also holds true for MCs and dTCs individually where we find an average distance to the respective MC midpoint of $135.4 \pm 91.3 \mu\text{m}$ vs. $93.4 \pm 57.5 \mu\text{m}$ (n.s., $p=0.22$, CI $-110.2;26.2$) and a distance to the respective dTC midpoint of $73.1 \pm 0 \mu\text{m}$ vs. $53.3 \pm 34.0 \mu\text{m}$ (n.s., $p=0.43$, CI $-72.21;32.6$). However, the sample size here is rather small so we refrain from going into more in-depth geometrical analysis although the principal polarization axis of the MCs seems different to the one we found in the MOR174-9 glomerulus.

3) The clustering of TC axons onto mitral cell bodies and is proposed as strongly indicative of TC-GC-MC connections, but no direct evidence is given to support this proposed interpretation. Given that this result is presented as a major finding of the manuscript, as the anatomical basis of a novel parallel circuit, additional evidence is needed to support the claim.

Thank you very much for pointing out this aspect. The clustering of TC axons beneath MC somata of the same glomerulus is indeed one of the findings we make when applying the technique we present to the OB. It follows on from the development of the technique, its validation and the analysis of dendritic stratification, soma location etc. We did not want to overstate this result but present it as an example how sparse higher order anatomical features (that are very difficult to obtain with other techniques) could be delineated with the new electroporation technique. We now have largely followed the suggestion of reviewer 3 and deemphasized this aspect.

Nevertheless, we have attempted to obtain additional experimental evidence for the proposed circuit. Presenting additional, conclusive, direct evidence for the speculative circuit is hampered by several experimental obstacles. Firstly, obtaining electrophysiological recordings from connected MC-GCs is very difficult (the only published recordings to our knowledge are a small number of VC recordings by Isaacson without morphological verification (38); the difficulty was summarized by Egger et al (39) "we first sought to examine the effect ... using paired recordings of granule and mitral cells ($n > 100$). However, this approach was precluded by an extremely low success rate in finding connected pairs") and we as well failed to do so with >50 paired recording attempts. Finding a third TC connected to both we would consider virtually impossible. In the past we had thus attempted Rabies tracing (e.g. our previous work (40)) to increase these odds but unfortunately Rabies seems to not label GC-MC connections (Fig R10).

Figure S3. Modified RABV-Mediated Transsynaptic Tracing of GC, Related to Figure 2

(A) Confocal mosaic image (EGFP channel) of a coronal section through the OB of an adult mouse 7 days after stereotaxic delivery of the tracer virus cocktail (rAAVs/modified RABV) into the GCL. The syringe containing the virus cocktail (green) points at the injection site. Bright source GCs can be detected in the GCL at the injection site. Directly connected, presynaptic neurons are confined to the EPL. **Note the absence of EGFP-labeled MCs in the MCL, suggesting that the RABV cannot cross dendrodendritic, reciprocal synapses.**
 (B) Magnification of the region in (A) outlined by the white dotted rectangle demonstrating the absence of RABV-labeled, EGFP-positive MCs. White dotted lines demarcate the EPL, OB, olfactory bulb; GCL, granule cell layer; MCL, mitral cell layer; EPL, external plexiform layer; GL, glomerular layer.

Figure S7. RABV-Mediated Transsynaptic Tracing of Presynaptic Inputs onto Single MCs, Related to Figure 6

(A) Schematic of the head of a newborn (P0) mouse injected bilaterally into the lateral ventricles with an rAAV expressing RG under control of the synapsin promoter (rAAV8 syn-RABVGlyco). Injection results in selective labeling of MCs in the OB (left).
 (B) Schematic of a sagittal section through an adult mouse brain injected at P0 with rAAV8 syn-RABVGlyco and subsequently injected into the PC with a RABV expressing EGFP (RABVΔG-EGFP) (green syringe). Triangles in OB represent MCs. Green dots in OB represent monosynaptically connected interneurons in EPL. Green lines indicate MC axons projecting into PC (green ellipse).
 (C) Confocal image of a RABV-infected, rAAV8 syn-RABVGlyco expressing MC (EGFP channel) showing monosynaptically connected interneurons in EPL. White arrow points at MCs. White ellipses indicate the position of glomeruli.
 (D) Another example of single MC connectivity maps as described in (B).
Note that labeling of GC is absent, suggesting that the RABV cannot cross dendrodendritic, reciprocal synapses. OB, olfactory bulb; CTX, cerebral cortex; HP, hippocampus; PC, piriform cortex; MC, mitral cells; MCL, mitral cell layer; EPL, external plexiform layer; GL, glomerular layer.

Figure R10 Rabies tracing attempts from GCs and MCs

Both figures and legends reproduced from Niedworok et al Cell Reports (3)

The key problem highlighted by these failed approaches is the main strength of the volume electroporation technique namely the ability to comprehensively label a defined volume and group of neurons associated with it through somata, dendrites or possibly axons, yet label them in full to allow investigation of their neurites in distant regions (MC and TC and their respective axons several 100 μm away from the site of electroporation). No other technique to our knowledge allows to draw conclusions about these sparse long-range circuits.

To get some additional evidence for the proposed circuit, we thus performed volume electron microscopic reconstruction of a region from EPL to IPL. Unfortunately, current volume EM techniques do not yet allow to reconstruct the entire glomerulus-GCL circuitry (which was one of the key reasons for us to develop the volume electroporation technique).

We thus performed electron microscopic analysis on a smaller volume (Stack 1 - 180 μm x 180 μm x 40 μm at a resolution of 13 nm x 13 nm x 32 nm, Fig R12a) using serial block-face scanning EM, encompassing parts of the EPL, MCL and IPL with the aim to identify individual GCs, synapses onto GC dendrites and follow presynaptic axons for at least sufficient length to identify whether they would stem from EPL and thus likely from superficial TCs. From a separate EM experiment (Stack 2 - 180 μm x 180 μm x 200 μm at the same resolution, Fig R11a) we could identify axons from sTCs as myelinated axons that traverse the EPL near perpendicular to the MCL/GL (Fig R11). In Stack 1 in turn we identified such axons in the EPL portion as putative sTC axons. Following one of those axons (retraced by 2 tracers independently) across the MCL into the IPL we identified an axodendritic synapse (Fig R12 c) onto a postsynaptic cell in the IPL. In turn reconstructing this cell showed that it possessed a soma in the GCL and spiny dendrites, thus corresponding to the classical GC morphology. Unfortunately, due to the size limit of the stack at hand, we could not follow this GC into the MCL/EPL. We therefore reconstructed an additional GC (starting with a GCL soma, Fig R12d) that was comprised in the imaged volume. Indeed, upon entering the MCL this GC made a dendro-somatic synapse onto a MCL cell (Fig R12d, e).

Figure R11 EM identification of TC axons (Supplementary Fig. 7 of the revised manuscript)

Tufted Cells display myelinated axons that cross the EPL. SBEM dataset fully containing a glomerulus and adjacent EPL is shown in (a). Tufted cell neurons (b) were identified based on their characteristic cytoarchitectural features, namely a pale cytoplasm, a single apical dendrite that branches profusely in one glomerulus, and the presence of long-range lateral dendrites (asterisk). In some cases an axon was easily identifiable (arrowhead). This axon eventually became myelinated (box in b reported in (c)). Abbreviations: ONL, olfactory nerve layer; GL, glomerular layer; EPL, external plexiform layer.

We have thus identified tentative examples of a tufted cell axon->GC (axodendritic) synapse as well as a potentially powerful GC->MC dendro-somatic synapse, the two ingredients in the proposed TC->GC->MC circuitry.

We realize that this is again indirect evidence for the proposed circuitry as we couldn't show unequivocally the source of the axons impinging onto GC dendrites and are unable to assess the glomerular affiliations of putative TC and MC. We thus decided to present this additional evidence only as supplementary figures 7 and 8, mention and discuss them on p. 9 and 12 and generally deemphasize this aspect of the paper (e.g. p. 9, abstract).

Figure R12 EM identification of putative TC->GC axodendritic and GC->MC dendro-somatic synapses (Supplementary Fig. 8 of the revised manuscript)

EPL-originated myelinated axons establish synapses in the IPL onto GC. SBEM dataset containing layers EPL, MCL, IPL and GCL (a). Myelinated axons in the EPL were identified and traced as they entered the IPL (b). These axons were found to establish axodendritic synapses (box in b reported in (c)) onto a granule cell (yellow). This dataset contained other granule cells ((d), pink) that established synapses onto MC (box in d reported in (e), MC in green). Abbreviations: EPL, external plexiform layer; MCL, mitral cell layer; IPL, internal plexiform layer; GCL, granule cell layer; MC, mitral cell; GC, granule cell.

Reviewer #3 (Remarks to the Author):

In the manuscript by D. Schwarz et al., the authors describe a novel method using nano-engineered electroporation microelectrodes to label neuronal networks associated with single olfactory glomeruli. They show that this new method allows for improved current density, improved current distribution, and electroporation efficacy by allowing a greater volume of material to be electroporated through multiple distributed release sites, with less damage than previously described local electroporation approaches. The authors applied this technique to characterize the neurons associated with a single glomerulus, quantifying the associated cell-types by allocating them into different layers of the olfactory bulb, and further characterizing the projection patterns of the labeled cells. Additionally, from these data the authors further propose a novel feedforward circuit from tufted cells that ultimately affects mitral cells. Overall, this paper presents a novel technique for the study of isolated neuronal networks that could be broadly applied for the detailed study of brain circuits. The technique seems to be very robust, and the manuscript is a great fit for the journal by addressing a few points.

We want to thank the reviewer for their detailed summary and insightful comments and valuable suggestions that we have addressed with additional data, analysis and editing of the revised text as detailed below.

Comments:

1. It would be useful to include data, or at least describe differences in electroporation volume, and/or labeling efficiency using different patterns or numbers of release sites. This would nicely augment the modelling data.

We thank the reviewer for this comment and suggestion and now included other comparative examples of FIB-produced pipette designs and corresponding FEM models as supplementary figure 2 referenced on page 5 from which important principles can be derived of how the electric potential distribution changes depending on number and location of additional release sites (Figure R13a). In the cases shown, we added two release levels at 10 and 20 μm from the tip with an edge length of 3 μm and 2.8 μm , respectively; in one case only from two sides and in the other from all four sides. As expected, the overall volume beyond 700 mV is reduced by adding additional release sites proximal to the pipette tip (2x2: 115.7 μm^3 and 2x4: 60.9 μm^3 compared to 220.1 μm^3 in the case of a normal pipette and 20.4 μm^3 in the case of the 5x4 design) as is the maximum transmembrane potential which is reached (2x2: 2.3 V, 2x4: 1.3 V compared to 8.99 V in the case of a normal pipette and 1.88 V in the case of the 5x4 design). In the latter case (2x4 design), maximum transmembrane potential is even lower than in the case of the 5x4 design since large amounts (in our case 44.57 $\mu\text{A}/50 \mu\text{A}$ = almost 90%) of the total current are already released at the proximal hole level and distributed over a large hole surface. Overall electroporation volume (>200 mV), however, is more difficult to predict: The overall effective electroporation volume in the 2x2 condition is 9195 μm^3 and thus slightly lower, in the case of the 4x2 design, which is 8430 μm^3 , even strongly reduced compared to our 5x4 design (9546 μm^3). The fact that this volume is also substantially reduced in the 4x2 case compared to the 2x2 design shows that distributing additional electroporation sites (and thus the current) available along the pipette principal axis is more effective in increasing electroporation volume than around the circular circumference. It is therefore important to “spare” as much current as possible in this current-divider circuit towards the pipette tip by an appropriately decreasing hole size as in the case of our 5x4 design. The optimal distribution of the electric field for electroporation certainly depends on the exact position, shape and size of inserted release sites so that a true optimization is very difficult to achieve due to the large number of free parameters.

In Figure R13b we show that FIB-fabrication itself is not a limiting factor to the shapes of additional release sites around the pipette tip as even ‘exotic’ shapes like a triangle (which may seem like the ultimate solution to the constraints mentioned above) can easily be constructed. However, such large openings with long edges can result in increased mechanical damage when inserted into biological tissue: Figure R13biv shows how large parts of a GFP expressing glomerulus are removed when pulling out the pipette from an acute brain slice.

Figure R13 Examples of other pipette designs and numerical modelling (Supplementary fig. 2 of the revised manuscript)

(a) Examples of a pipette featuring a 2x2 and 2x4 design. (i) Example of an NEM after successful insertion of the two-level hole design at 10 and 20 μm from the tip with an edge length of 3 μm and 2.8 μm , respectively, as seen in high-resolution FIB imaging mode. (ii) Corresponding top view in the SEM mode. (iii) Corresponding cross-section of the 3D-FEM model illustrating total effective electroporation volume and its distribution around the pipette tip at 50 μA employing the 2x2 design. (iv) Same representation for the case of two release levels and holes from all four sides. **(b)** Example of physical damage when employing large release sites. (i) Example of an NEM after successful insertion of a triangular hole design extending from 2 μm above the tip to 20 μm proximal from the tip, as seen in high-resolution FIB imaging mode. (ii) Corresponding top view in the SEM mode. (iii) light-microscopic image showing the inserted release site prior to insertion into an acute brain slice. (iv) green fluorescent microscopic image showing the pipette tip after retraction from the MOR174-GFP glomerulus in an acute brain slice removing large parts of the glomerular tissue.

2. Although it is appreciated that a focus of the analysis was to categorize projection neurons to glomeruli, the olfactory bulb layers contain heterogeneous cell populations. It would be informative to include some marker analysis to definitively identify TCs vs. MCs vs. SMCs vs. interneurons. The criteria are not always clear in the text of how the authors parse out the different projection neuron subtypes aside from anatomy.

Here the reviewer raises a very important point, namely how to define different cell types. We have now included a clearer definition of cell types in the revised manuscript on page 7.

In brief, in our understanding, the gold standard criterion for distinguishing MC vs TC cells is indeed morphology (e.g. Haberly & Price 1977, Mori et al 1983, Orona et al 1984, our own previous work Fukunaga et al. 2012 (41-44)). Regarding specific markers, the best marker we are aware of to label MCs (tbet / tbx21 e.g. Haddad et al. (45)) is indeed labeling all varieties of TCs as well (Figure R14 from our own experiments) so we cannot take such a marker as differentiator.

Figure R 14 Overview confocal image of *tbet/tbx21* expression in the olfactory bulb.

AAV-flex-GCaMP6f was injected into the olfactory bulb of a *tbet-Cre* mouse. Green channel shows *GCaMP6f* fluorescence, Blue channel nuclear DAPI staining

While the presence of lateral dendrites should anatomically rule out any potential confusion between PGCs and TCs in the GL (see e.g. Fig R6), we now complemented glomerular electroporation with post-hoc immunostaining of Calbindin (CB) (Figure R15a). CB is known to almost exclusively stain interneurons of the GL unlike other markers like Tyrosin-Hydroxylase or neurocalcin which are also expressed by tufted cells (15, 46, 47). In this experiment we do not find a single electroporated cell, that was identified as a TC based on morphological criteria, with positivity for CB, confirming the validity of the morphological definition.

The other region where MCs/TCs could be most readily confused with interneurons would be somata located in the upper EPL (e.g. van Gehuchten neurons or short axon cells). Such neurons can be defined by their

positive staining of parvalbumin (PV) (48). We therefore stained for PV and compared these cells to electroporated TCs. Indeed, PV-positive neurons of the EPL did not exhibit any lateral dendrites comparable to our TCs and these neurons did not possess clear apical dendrites nor did they extend any processes into the GL (Fig. R15b). Instead, these neurons gave rise to fine neuritic processes which contributed to an intricate meshwork within the EPL. Given that all TCs which were labelled by electroporation possessed (by our definition) apical and lateral dendrites and not a single of these was PV-positive, we conclude that the morphological classification accurately captures projection neurons.

Figure R15 Complementary immunohistochemistry

(a) Complementary post-hoc immunohistochemistry following electroporation. Upper panel showing CB-positive neurons within and around the GL (left) and a representative corresponding plane of the TMR-channel under the confocal microscope (right). Lower panel shows an overlay image in which superficial TCs are clearly distinct from adjacent CB-positive neurons without double labelling. **(b)** PV-positive interneurons in the upper EPL which do not extend any processes into the GL. Scale always 50 μm .

3. Given the claim “this association strongly suggests a feedforward TC to GC to MC pathway”, it now seems almost imperative for provide more than anatomical analysis to provide cell type specificity. The manuscript would benefit from either building upon this speculation with data, or deemphasizing (or removing) this point.

Thank you for pointing this out. As the reviewer suggests, the proposed feedforward circuitry is an interesting hypothesis we could generate based on the electroporation data, however not a central part of the manuscript. We have thus – as suggested – deemphasized this point to not potentially provide an undue focus of the manuscript. Nevertheless, we have investigated several approaches how we could further investigate this hypothesis. Here we re-iterate our response to reviewer 2 (above) who raises a similar point:

Presenting additional, conclusive, direct evidence for the speculative circuit is hampered by several experimental obstacles. Firstly, obtaining electrophysiological recordings from connected MC-GCs is very difficult (the only published recordings to our knowledge are a small number of VC recordings by Isaacson without morphological verification (38); the difficulty was summarized by Egger et al (39) “we first sought to examine the effect ... using paired recordings of granule and mitral cells (n > 100). However, this approach was precluded by an extremely low success rate in finding connected pairs”) and we as well failed to do so with >50 paired recording attempts. Finding a third TC connected to both we would consider virtually impossible. In the past we had thus attempted Rabies tracing (e.g. our previous work (40)) to increase these odds but unfortunately Rabies seems to not label GC-MC connections (Fig R10).

Figure S3. Modified RABV-Mediated Transsynaptic Tracing of GC, Related to Figure 2

(A) Confocal mosaic image (EGFP channel) of a coronal section through the OB of an adult mouse 7 days after stereotaxic delivery of the tracer virus cocktail (rAAVs/modified RABV) into the GCL. The syringe containing the virus cocktail (green) points at the injection site. Bright source GCs can be detected in the GCL at the injection site. Directly connected, presynaptic neurons are confined to the EPL. **Note the absence of EGFP-labeled MCs in the MCL, suggesting that the RABV cannot cross dendrodendritic, reciprocal synapses.**
 (B) Magnification of the region in (A) outlined by the white dotted rectangle demonstrating the absence of RABV-labeled, EGFP-positive MCs. White dotted lines demarcate the EPL. OB, olfactory bulb; GCL, granule cell layer; MCL, mitral cell layer; EPL, external plexiform layer; GL, glomerular layer.

Figure S7. RABV-Mediated Transsynaptic Tracing of Presynaptic Inputs onto Single MCs, Related to Figure 6

(A) Schematic of the head of a newborn (P0) mouse injected bilaterally into the lateral ventricles with an rAAV expressing RG under control of the synapsin promoter (rAAV8 syn-RABVGlyco). Injection results in selective labeling of MCs in the OB (left).
 (B) Schematic of a sagittal section through an adult mouse brain injected at P0 with rAAV8 syn-RABVGlyco and subsequently injected into the PC with a RABV expressing EGFP (RABVΔG-EGFP) (green syringe). Triangles in OB represent MCs. Green dots in OB represent monosynaptically connected interneurons in EPL. Green lines indicate MC axons projecting into PC (green ellipse).
 (C) Confocal image of a RABV-infected, rAAV8 syn-RABVGlyco expressing MC (EGFP channel) showing monosynaptically connected interneurons in EPL. White arrow points at MCs. White ellipses indicate the position of glomeruli.
 (D) Another example of single MC connectivity maps as described in (B).
Note that labeling of GC is absent, suggesting that the RABV cannot cross dendrodendritic, reciprocal synapses. OB, olfactory bulb; CTX, cerebral cortex; HP, hippocampus; PC, piriform cortex; MC, mitral cells; MCL, mitral cell layer; EPL, external plexiform layer; GL, glomerular layer.

Figure R10 (repeated from response to Reviewer 2) Rabies tracing attempts from GCs and MCs

Both figures and legends reproduced from Niedworok et al Cell Reports (3)

The key problem highlighted by these failed approaches is the main strength of the volume electroporation technique namely the ability to comprehensively label a defined volume and group of neurons associated with it through somata, dendrites or possibly axons, yet label them in full to allow investigation of their neurites in distant regions (MC and TC and their respective axons several 100 μm away from the site of electroporation). No other technique to our knowledge allows to draw conclusions about these sparse long-range circuits.

To get some additional evidence for the proposed circuit, we thus performed volume electron microscopic reconstruction of a region from EPL to IPL. Unfortunately, current volume EM techniques do not yet allow to reconstruct the entire glomerulus-GCL circuitry (which was one of the key reasons for us to develop the volume electroporation technique).

We thus performed electron microscopic analysis on a smaller volume (Stack 1 - 180 μm x 180 μm x 40 μm at a resolution of 13 nm x 13 nm x 32 nm, Fig R12a) using serial block-face scanning EM, encompassing parts of the EPL, MCL and IPL with the aim to identify individual GCs, synapses onto GC dendrites and follow presynaptic axons for at least sufficient length to identify whether they would stem from EPL and thus likely from superficial TCs. From a separate EM experiment (Stack 2 - 180 μm x 180 μm x 200 μm at the same resolution, Fig R11a) we could identify axons from sTCs as myelinated axons that traverse the EPL near perpendicular to the MCL/GL (Fig R11). In Stack 1 in turn we identified such axons in the EPL portion as putative sTC axons. Following one of those

Figure R11(repeated from response to Reviewer 2) EM identification of TC axons (re-iterated)

Tufted Cells display myelinated axons that cross the EPL. SBEM dataset fully containing a glomerulus and adjacent EPL is shown in (a). Tufted cell neurons (b) were identified based on their characteristic cytoarchitectural features, namely a pale cytoplasm, a single apical dendrite that branches profusely in one glomerulus, and the presence of long-range lateral dendrites (asterisk). In some cases an axon was easily identifiable (arrowhead). This axon eventually became myelinated (box in b reported in (c)). Abbreviations: ONL, olfactory nerve layer; GL, glomerular layer; EPL, external plexiform layer.

axons (retraced by 2 tracers independently) across the MCL into the IPL we identified an axodendritic synapse (Fig R12 c) onto a postsynaptic cell in the IPL. In turn reconstructing this cell showed that it possessed a soma in the GCL and spiny dendrites, thus corresponding to the classical GC morphology. Unfortunately, due to the size limit of the stack at hand, we could not follow this GC into the MCL/EPL. We therefore reconstructed an additional GC (starting with a GCL soma, Fig R12d) that was comprised in the imaged volume. Indeed, upon entering the MCL this GC made a dendro-somatic synapse onto a MCL cell (Fig R12d, e).

We have thus identified tentative examples of a tufted cell axon->GC (axodendritic) synapse as well as a potentially powerful GC->MC dendro-somatic synapse, the two ingredients in the proposed TC->GC->MC circuitry.

We realize that this is again indirect evidence for the proposed circuitry as we couldn't show unequivocally the source of the axons impinging onto GC dendrites and are unable to assess the glomerular affiliations of putative TC and MC. We thus decided to present this additional evidence only as supplementary figures 7 and 8, mention and discuss them on p. 9 and 13 and deemphasize this aspect of the paper as suggested (e.g. p. 9).

Figure R12 (repeated from response to Reviewer 2) EM identification of putative TC->GC axodendritic and GC->MC dendro-somatic synapses (re-iterated)

EPL-originated myelinated axons establish synapses in the IPL onto GC. SBEM dataset containing layers EPL, MCL, IPL and GCL (a). Myelinated axons in the EPL were identified and traced as they entered the IPL (b). These axons were found to establish axodendritic synapses (box in b reported in (c)) onto a granule cell (yellow). This dataset contained other granule cells ((d), pink) that established synapses onto MC (box in d reported in (e), MC in green). Abbreviations: EPL, external plexiform layer; MCL, mitral cell layer; IPL, internal plexiform layer; GCL, granule cell layer; MC, mitral cell; GC, granule cell.

4. Practical discussion of other applications using this platform would be welcomed. For example, does this approach work to electroporate local neurons with DNA constructs?

This is indeed an excellent point and we discuss potential other application in more detail on p 13 the revised manuscript.

5. The supplementary figures are listed out of order throughout the text. These should be chronologically labeled as the figure data is described.

We apologize and have corrected the order in the revised manuscript.

References:

1. Kiyokage E, Pan YZ, Shao ZY, et al. Molecular Identity of Periglomerular and Short Axon Cells. *Journal of Neuroscience*. 2010;30(3):1185-96.
2. Kosaka T, Kosaka K. "Interneurons" in the olfactory bulb revisited. *Neurosci Res*. 2011;69(2):93-9.
3. Niedworok CJ, Schwarz I, Ledderose J, et al. Charting monosynaptic connectivity maps by two-color light-sheet fluorescence microscopy. *Cell Rep*. 2012;2(5):1375-86.
4. Kiyokage E, Pan YZ, Shao ZY, et al. Molecular Identity of Periglomerular and Short Axon Cells. *J Neurosci*. 2010;30(3):1185-96.
5. Kosaka T, Kosaka K. Neuronal organization of the main olfactory bulb revisited. *Anat Sci Int*. 2016;91(2):115-27.
6. Towhidi L, Kotnik T, Pucihar G, et al. Variability of the Minimal Transmembrane Voltage Resulting in Detectable Membrane Electroporation. *Electromagn Biol Med*. 2008;27(4):372-85.
7. Henslee BE, Morss A, Hu X, et al. Electroporation Dependence on Cell Size: Optical Tweezers Study. *Anal Chem*. 2011;83(11):3998-4003.
8. Sale AJH, Hamilton WA. Effects of High Electric Fields on Micro-Organisms .3. Lysis of Erythrocytes and Protoplasts. *Biochim Biophys Acta*. 1968;163(1):37-43.
9. Royet JP, Distel H, Hudson R, Gervais R. A re-estimation of the number of glomeruli and mitral cells in the olfactory bulb of rabbit. *Brain Res*. 1998;788(1-2):35-42.
10. Royet JP, Souchier C, Jourdan F, Ploye H. Morphometric Study of the Glomerular Population in the Mouse Olfactory-Bulb - Numerical Density and Size Distribution Along the Rostrocaudal Axis. *J Comp Neurol*. 1988;270(4):559-68.
11. Royet JP, Jourdan F, Ploye H, Souchier C. Morphometric Modifications Associated with Early Sensory Experience in the Rat Olfactory-Bulb .2. Stereological Study of the Population of Olfactory Glomeruli. *J Comp Neurol*. 1989;289(4):594-609.
12. Pomeroy SL, LaMantia AS, Purves D. Postnatal construction of neural circuitry in the mouse olfactory bulb. *J Neurosci*. 1990;10(6):1952-66.
13. Allison AC, Warwick RT. Quantitative observations on the olfactory system of the rabbit. *Brain*. 1949;72(Pt. 2):186-97.
14. Richard MB, Taylor SR, Greer CA. Age-induced disruption of selective olfactory bulb synaptic circuits. *Proc Natl Acad Sci U S A*. 2010;107(35):15613-8.
15. Parrish-Aungst S, Shipley MT, Erdelyi F, et al. Quantitative analysis of neuronal diversity in the mouse olfactory bulb. *J Comp Neurol*. 2007;501(6):825-36.

16. Benson TE, Ryugo DK, Hinds JW. Effects of sensory deprivation on the developing mouse olfactory system: a light and electron microscopic, morphometric analysis. *J Neurosci.* 1984;4(3):638-53.
17. Nawroth JC, Greer CA, Chen WR, et al. An energy budget for the olfactory glomerulus. *J Neurosci.* 2007;27(36):9790-800.
18. Meisami E, Safari L. A quantitative study of the effects of early unilateral olfactory deprivation on the number and distribution of mitral and tufted cells and of glomeruli in the rat olfactory bulb. *Brain Res.* 1981;221(1):81-107.
19. Panhuber H, Laing DG, Willcox ME, et al. The distribution of the size and number of mitral cells in the olfactory bulb of the rat. *J Anat.* 1985;140 (Pt 2):297-308.
20. Liu A, Savya S, Urban NN. Early Odorant Exposure Increases the Number of Mitral and Tufted Cells Associated with a Single Glomerulus. *J Neurosci.* 2016;36(46):11646-53.
21. Sosulski DL, Bloom ML, Cutforth T, et al. Distinct representations of olfactory information in different cortical centres. *Nature.* 2011;472(7342):213-6.
22. Ke MT, Imai T. Optical clearing of fixed brain samples using SeeDB. *Curr Protoc Neurosci.* 2014;66:Unit 2 22.
23. Carleton A, Petreanu LT, Lansford R, et al. Becoming a new neuron in the adult olfactory bulb. *Nat Neurosci.* 2003;6(5):507-18.
24. Mizrahi A, Lu J, Irving R, et al. In vivo imaging of juxtglomerular neuron turnover in the mouse olfactory bulb. *Proc Natl Acad Sci U S A.* 2006;103(6):1912-7.
25. Adam Y, Mizrahi A. Long-term imaging reveals dynamic changes in the neuronal composition of the glomerular layer. *J Neurosci.* 2011;31(22):7967-73.
26. Alonso M, Ortega-Perez I, Grubb MS, et al. Turning astrocytes from the rostral migratory stream into neurons: a role for the olfactory sensory organ. *J Neurosci.* 2008;28(43):11089-102.
27. Alonso M, Viollet C, Gabellec MM, et al. Olfactory discrimination learning increases the survival of adult-born neurons in the olfactory bulb. *J Neurosci.* 2006;26(41):10508-13.
28. Zapiec B, Mombaerts P. Multiplex assessment of the positions of odorant receptor-specific glomeruli in the mouse olfactory bulb by serial two-photon tomography. *Proc Natl Acad Sci U S A.* 2015;112(43):E5873-82.
29. Economo MN, Hansen KR, Wachowiak M. Control of Mitral/Tufted Cell Output by Selective Inhibition among Olfactory Bulb Glomeruli. *Neuron.* 2016;91(2):397-411.
30. Willhite DC, Nguyen KT, Masurkar AV, et al. Columnar organization in the olfactory bulb. *Chemical Senses.* 2006;31(5):A75-A.
31. Willhite DC, Nguyen KT, Masurkar AV, et al. Viral tracing identifies distributed columnar organization in the olfactory bulb. *Proc Natl Acad Sci U S A.* 2006;103(33):12592-7.
32. Kim DH, Phillips ME, Chang AY, et al. Lateral connectivity in the olfactory bulb is sparse and segregated. *Front Neural Circuits.* 2011;5:5.
33. Luo M, Katz LC. Response Correlation Maps of Neurons in the Mammalian Olfactory Bulb. *Neuron.* 2001;32(6):1165-79.
34. Buonviso N, Chaput MA, Scott JW. Mitral cell-to-glomerulus connectivity: an HRP study of the orientation of mitral cell apical dendrites. *J Comp Neurol.* 1991;307(1):57-64.
35. Mori K, Takahashi YK, Igarashi KM, Yamaguchi M. Maps of odorant molecular features in the mammalian olfactory bulb. *Physiol Rev.* 2006;86(2):409-33.
36. Kobayakawa K, Kobayakawa R, Matsumoto H, et al. Innate versus learned odour processing in the mouse olfactory bulb. *Nature.* 2007;450(7169):503-8.
37. Mori K, Matsumoto H, Tsuno Y, Igarashi KM. Dendrodendritic Synapses and Functional Compartmentalization in the Olfactory Bulb. In: Finger TE, ed. *International Symposium on Olfaction and Taste*; 2009:255-8.
38. Isaacson JS. Mechanisms governing dendritic gamma-aminobutyric acid (GABA) release in the rat olfactory bulb. *Proc Natl Acad Sci U S A.* 2001;98(1):337-42.

39. Egger V, Svoboda K, Mainen ZF. Mechanisms of lateral inhibition in the olfactory bulb: efficiency and modulation of spike-evoked calcium influx into granule cells. *J Neurosci.* 2003;23(20):7551-8.
40. Rancz EA, Franks KM, Schwarz MK, et al. Transfection via whole-cell recording in vivo: bridging single-cell physiology, genetics and connectomics. *Nat Neurosci.* 2011;14(4):527-32.
41. Haberly LB, Price JL. The axonal projection patterns of the mitral and tufted cells of the olfactory bulb in the rat. *Brain Res.* 1977;129(1):152-7.
42. Mori K, Kishi K, Ojima H. Distribution of dendrites of mitral, displaced mitral, tufted, and granule cells in the rabbit olfactory bulb. *J Comp Neurol.* 1983;219(3):339-55.
43. Orona E, Rainer EC, Scott JW. Dendritic and Axonal Organization of Mitral and Tufted Cells in the Rat Olfactory-Bulb. *J Comp Neurol.* 1984;226(3):346-56.
44. Fukunaga I, Berning M, Kollo M, et al. Two distinct channels of olfactory bulb output. *Neuron.* 2012;75(2):320-9.
45. Haddad R, Lanjuin A, Madisen L, et al. Olfactory cortical neurons read out a relative time code in the olfactory bulb. *Nat Neurosci.* 2013;16(7):949-57.
46. Kosaka T, Kosaka K. Heterogeneity of calbindin-containing neurons in the mouse main olfactory bulb: I. General description. *Neuroscience Research.* 2010;67(4):275-92.
47. Davis BJ, Macrides F. Tyrosine-Hydroxylase Immunoreactive Neurons and Fibers in the Olfactory System of the Hamster. *Journal of Comparative Neurology.* 1983;214(4):427-40.
48. Kosaka T, Kosaka K. Heterogeneity of parvalbumin-containing neurons in the mouse main olfactory bulb, with special reference to short-axon cells and betaIV-spectrin positive dendritic segments. *Neurosci Res.* 2008;60(1):56-72.

REVIEWERS' COMMENTS:

Reviewer #1 (Remarks to the Author):

The authors have thoroughly addressed all my previous concerns.

Reviewer #2 (Remarks to the Author):

I would like to commend the authors for addressing thoroughly the first two points I raised through numerous additional experiments and elegant analyses. The new data and panels presented strengthen significantly the manuscript. I am also in agreement with the authors that probing the validity of the novel proposed inhibitory circuit necessitates additional evidence. This is currently lacking. To my mind, the new data presented is suggestive, but remains an indirect piece of evidence: the presence on synaptic connections between putative TC collateral axons and dendrites of IPC GCs and between MC and other different IPL GCs does not add direct evidence for a TC-GC-MC inhibitory neuronal circuit. In my opinion, the revised manuscript can stand on its own in Nature Communications as a useful technological advance, without making specific circuit claims that are not fully supported. While it is important to point out the proximity of putative TC collateral axons to the sister MCs somas, and discuss the potential circuit implications, I still recommend that the authors remove the statements related to the putative TC-GC-MC pathway from the abstract, further downplay them in the Discussion, and perhaps make this the focus of a future study.

Reviewer #3 (Remarks to the Author):

The comprehensive set of revisions made by Schwarz et al., significantly strengthen the claims and quality of the revised manuscript. This approach is now very well described, and is supported by high quality data.

I fully endorse publication in Nature Communications.

Reviewer #1 (Remarks to the Author):

The authors have thoroughly addressed all my previous concerns.

We thank the reviewer for agreeing with our efforts to address the previous concerns helping us to strengthen our manuscript significantly.

Reviewer #2 (Remarks to the Author):

I would like to commend the authors for addressing thoroughly the first two points I raised through numerous additional experiments and elegant analyses. The new data and panels presented strengthen significantly the manuscript. I am also in agreement with the authors that probing the validity of the novel proposed inhibitory circuit necessitates additional evidence. This is currently lacking. To my mind, the new data presented is suggestive, but remains an indirect piece of evidence: the presence on synaptic connections between putative TC collateral axons and dendrites of IPC GCs and between MC and other different IPL GCs does not add direct evidence for a TC-GC-MC inhibitory neuronal circuit. In my opinion, the revised manuscript can stand on its own in Nature Communications as a useful technological advance, without making specific circuit claims that are not fully supported. While it is important to point out the proximity of putative TC collateral axons to the sister MCs somas, and discuss the potential circuit implications, I still recommend that the authors remove the statements related to the putative TC-GC-MC pathway from the abstract, further downplay them in the Discussion, and perhaps make this the focus of a future study.

We thank the reviewer for this positive and highly encouraging suggestion. As requested, we removed the statement from the abstract and softened our claims on p. 13 in the discussion section of the manuscript. We are currently indeed focusing on our TC-GC-MC findings in a separate study.

Reviewer #3 (Remarks to the Author):

The comprehensive set of revisions made by Schwarz et al., significantly strengthen the claims and quality of the revised manuscript. This approach is now very well described, and is supported by high quality data.

I fully endorse publication in Nature Communications.

We thank the reviewer for this very positive feedback on our revised manuscript.